# Mapping the dynamics of force transduction at cell–cell junctions of epithelial clusters

**Mei Rosa Ng†, Achim Besser†, Joan S Brugge, Gaudenz Danuser*‡**

Department of Cell Biology, Harvard Medical School, Boston, United States

**Abstract** Force transduction at cell-cell adhesions regulates tissue development, maintenance and adaptation. We developed computational and experimental approaches to quantify, with both sub-cellular and multi-cellular resolution, the dynamics of force transmission in cell clusters. Applying this technology to spontaneously-forming adherent epithelial cell clusters, we found that basal force fluctuations were coupled to E-cadherin localization at the level of individual cell-cell junctions. At the multi-cellular scale, cell-cell force exchange depended on the cell position within a cluster, and was adaptive to reconfigurations due to cell divisions or positional rearrangements. Importantly, force transmission through a cell required coordinated modulation of cell-matrix adhesion and actomyosin contractility in the cell and its neighbors. These data provide insights into mechanisms that could control mechanical stress homeostasis in dynamic epithelial tissues, and highlight our methods as a resource for the study of mechanotransduction in cell-cell adhesions.

***For correspondence:** gaudenz. danuser@utsouthwestern.edu

†These authors contributed equally to this work

**Present address:** ‡Department of Cell Biology, UT Southwestern Medical Center, Dallas, United States

**Competing interests:** The authors declare that no competing interests exist.

**Reviewing editor**: Ewa Paluch, University College London, United Kingdom

## Introduction

Tissues undergo continuous rearrangements that require tightly balanced exchanges of mechanical forces between individual cells through cadherin-mediated cell–cell junctions. The importance of mechanical coupling between cells has been documented for diverse multi-cellular processes, including tissue morphogenesis during development (*Mammoto and Ingber, 2010*; *Martin et al., 2010*; *Rauzi et al., 2010*; *Lecuit et al., 2011*; *Weber et al., 2012*), collective migration during wound healing (*Trepat et al., 2009*; *Tambe et al., 2011*; *Ng et al., 2012*), epithelial and endothelial barrier functions (*Tzima et al., 2005*; *Twiss et al., 2012*), and cancer progression (*Friedl et al., 2004*; *Bajpai et al., 2009*). In all these processes, the spatiotemporal regulation of mechanical forces is essential for the maintenance of tissue integrity as well as for communication and coordination between cells. Nevertheless, it is yet unclear how the magnitude and length scale of cell–cell force transmission are modulated dynamically as a function of cell position and topological changes within a cellular ensemble. Moreover, as recent studies have begun to characterize the molecular basis of the mechano-responsiveness of the cadherin–catenin complex (*Gomez et al., 2011*; *Leckband et al., 2011*), it is important to understand how the recruitment and regulation of cadherin molecules at the adherens junctions are coupled to cell–cell force transmission.

To address these questions, an assay is required that can measure cell–cell force transmission in situ and in *time*, which can be combined with high-resolution time-lapse imaging of protein recruitment and activity. We developed a method that utilizes high-resolution traction force microscopy (TFM) and time-lapse live cell microscopy to quantify cell–cell mechanical interactions in dynamic multi-cellular clusters, while maintaining the sub-cellular resolution needed to measure force exchange at the level of individual junctions. Our method expanded upon several pioneering studies, which inferred cell–cell force transmission from the regional imbalance of traction forces in cell pairs and three-cell clusters in linear configurations (*Liu et al., 2010*; *Maruthamuthu et al., 2011*; *McCain et al., 2012*; *Tseng et al., 2012*), by now providing the time and sub-cellular resolution needed to derive the spatiotemporal

**eLife digest** The intestines, liver, and skin are all examples of organs that perform specific functions. Organs are comprised of tissues, which are themselves made up of cells. Epithelial tissue is one of the four basic types of tissue found in animals, and it occurs in almost every organ in the body. For example, epithelial tissue makes up the outermost layer of the skin, and the lining of the lungs and the intestines; the cells in epithelial tissues are attached to one another via 'adhesion molecules'.

Organs and tissues need to be maintained throughout life in order for them to work properly. Epithelial cells in particular are very short-lived and must be constantly replaced. If epithelial tissue is cut or damaged in any way, the surrounding healthy epithelial cells must work together to repair the wound and restore the tissue's integrity. These processes require individual epithelial cells to communicate with one another. While chemical signals provide one means of cell-to-cell communication, cells also sense and respond to the physical presence of surrounding cells.

In adults, organs and tissues generally do not change shape or size; as such there is a tightly balanced exchange of mechanical forces between the individual cells. Damage to the tissue causes a detectable change in these mechanical forces, which is sensed by nearby healthy epithelial cells and causes them to work towards healing the wound. While the importance of mechanical forces in maintaining tissue integrity is widely recognized, there were few tools to study these forces; this meant that mechanical communication through cell–cell adhesion sites was not well understood.

Now Ng, Besser et al. describe the development and use of a new method for measuring and mapping the exchange of mechanical forces at cell–cell adhesion sites. Changes in the strength of the forces exchanged between cells could be measured across clusters of multiple cells or or specific parts of individual cells. Ng, Besser et al. found that when an epithelial cell in a cluster started to divide to form two new cells, the cell exerted less mechanical force on its neighboring cells.

Ng, Besser et al. found that the forces exerted between cells were strongest when there was more of an adhesion molecule called E-cadherin in the cell surface membrane at the cell–cell adhesion sites. The opposite was also true, as these forces were weakest at cell–cell adhesion sites with fewer E-cadherin molecules. The new method and findings will now help to guide future studies into how mechanical forces are transmitted between living cells.

coordination of force exchanges in larger cell groups as well as the coupling of force and protein dynamics at individual adherens junctions. To achieve reconstruction of forces at individual cell–cell junctions and in cell clusters of generic configurations, we incorporated the thin-plate modeling approach of cells and finite element analysis of the plate deformation that was utilized by other studies to infer intercellular interactions in entire cell sheets (*Trepat et al., 2009*; *Tambe et al., 2011*, *2013*; *Hur et al., 2012*).

Using this assay, we quantitatively mapped the forces transmitted through individual cell–cell junctions of dynamic MCF10A epithelial cell clusters of various sizes and configurations over time. Our results revealed that the spatial distributions and dynamics of basal cell–cell force transmission correlated with morphogenetic events such as cell divisions. Along the cell junction, we showed with sub-interface resolution that the local mechanical stresses also correlated with localization of E-cadherin molecules. Expanding our approach to temporal force fluctuation analysis at the length scale of a cell, we found that force propagation through cells can be intercepted by cell anchorage to the extracellular matrix and by actomyosin contraction. Together these results provide a first glimpse of the dynamics of sub-cellular force exchanges that yield stress homeostasis in quiescent and proliferating tissues.

## Results

### Implementation and extension of the force-balancing principle to calculate cell–cell forces in large adherent clusters

Our first step in implementing high-resolution quantification of cell–cell forces built on the force-balancing principle was introduced in previous studies (*Liu et al., 2010*; *Maruthamuthu et al., 2011*; *McCain et al., 2012*). The principle states that traction forces exerted by a single cell, or by a cluster of cells, are in mechanical equilibrium with the extracellular substrate. Hence, the traction forces integrated over the footprint of a single cell or a cell cluster must be equal to zero (*Figure 1*; see 'The

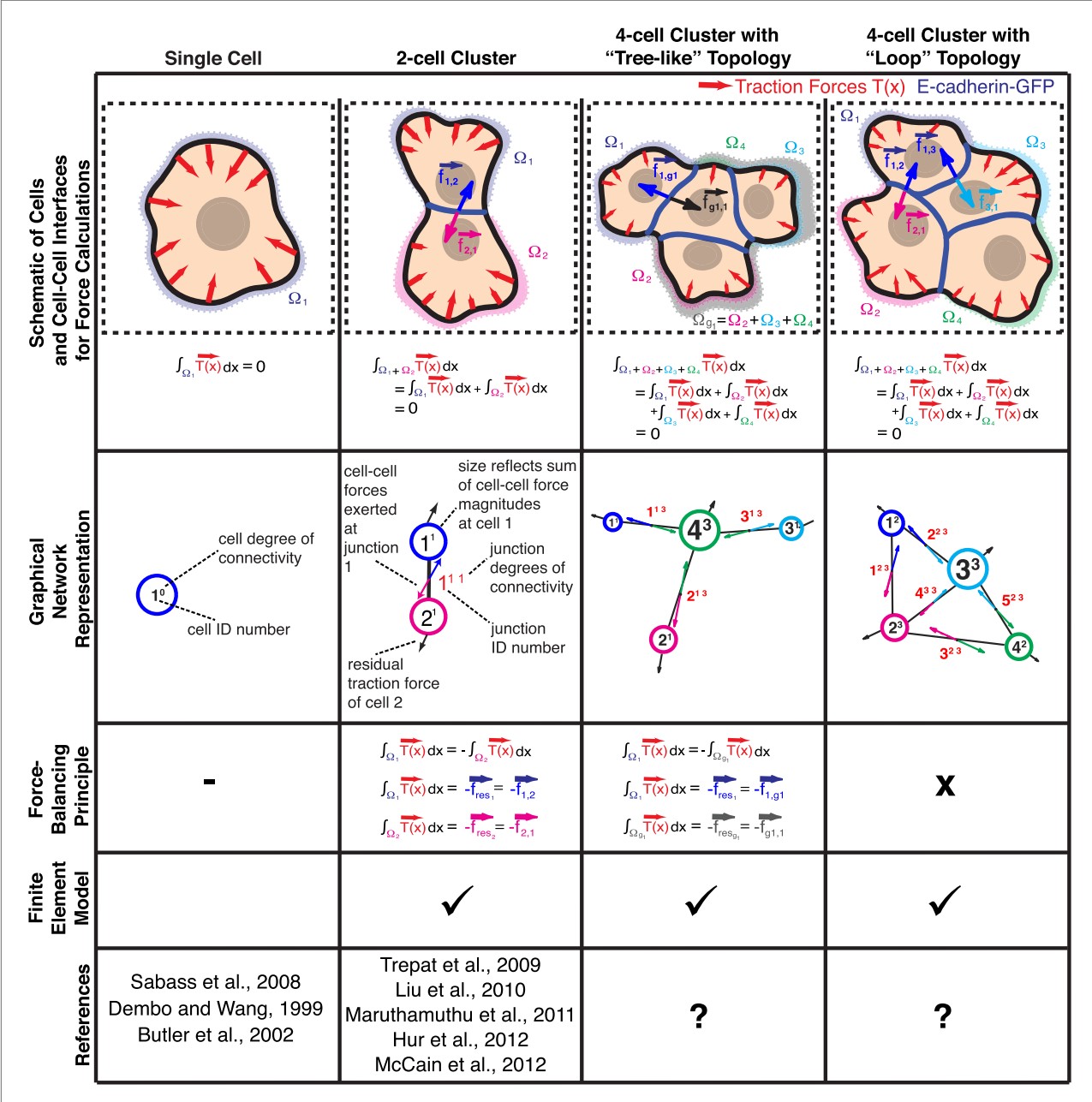

**Figure 1.** Calculation of cell–cell forces from traction forces. Single cells and cell clusters (depicted in cartoon and graphical network representations for up to four cells, with areas $\Omega_1$–$\Omega_4$) exert traction forces on the substrate (red vectors; $\overline{T}(\overline{x})$). Integration of traction forces over the footprint (color-shaded boundaries) of a single cell (column 1) or an entire cell cluster (columns 2–4) yields a balanced net force of 0. In cell pairs and clusters with a 'tree-like' topology (columns 2 and 3), forces exchanged at each cell–cell junction can be determined by partitioning the cluster into two sub-networks and calculating the residual force required to balance the traction forces integrated over the footprint of each sub-network. See 'Materials and methods' for details. In cell clusters with a 'loop' topology (column 4), individual cell–cell junctions do not fully partition the cluster into disjoint sub-networks. In such cases, force transmission within a cell cluster can be calculated based on a model that describes the cluster as a thin-plate in mechanical equilibrium with the traction forces. The corresponding stress distribution inside the thin-plate is computed by the finite element method (FEM).

force-balancing principle and its application to larger cell clusters' in 'Materials and methods' section). Following from this, if the integrated traction force over the footprint of an individual cell within a cell cluster is non-zero, other cells in the cluster must balance it by force transmission through cell–cell adhesions. Thus, the residual traction force for a particular cell within a cluster determines the vectorial sum of external forces this cell experiences through cell–cell adhesions (*Equation 1* in 'Materials and

methods'). In the case of a cell pair, each cell has one interface. Therefore, the residual forces associated with each of the two cells are—within the noise limits of traction force reconstruction—of equal magnitudes but opposite directions and indicate the force exchanged through that interface.

Whereas previous studies examined cell clusters with two or at most three cells in a linear configuration (*Liu et al., 2010*; *Maruthamuthu et al., 2011*; *McCain et al., 2012*), the force balancing principle readily expands to the calculation of cell–cell forces in larger cell clusters with linear or 'tree-like' topologies (*Figure 1*). To determine the neighborhood topology of cells, we defined a cell cluster as a graphical network, where each cell is represented as a node and each junction between two cells is represented by an edge connecting two nodes of the network. In a 'tree-like' topology, removal of any of the graphical edges results in two completely disjointed sub-networks. In this case, the forces at any cell–cell interface can be determined by calculating the residual forces over the cellular footprints defined by the two sub-networks (*Figure 1*; for details, see 'The force-balancing principle and its application to larger cell clusters' in 'Materials and methods' section). Statistical analyses of our experimental data indicate that the median error of force calculations based on the force-balancing principle was ~14% of the expected force magnitudes (*Figure 2*).

## Thin-plate model for finite element analysis of sub-cellular cell–cell force transmission in clusters of generic topology

Cell clusters may also adopt a 'loop' topology. In such 'loop' clusters, removal of single graphical edges no longer yields disjointed sub-networks (*Figure 1*). This means that cell–cell junctions are configured in a mechanically redundant system, and the forces exchanged at each junction cannot be simply resolved using the force-balancing principle. To resolve the relative contributions of redundant junctions to the force balance in cell clusters with a 'loop' topology, we modeled the cell cluster as a thin-plate in mechanical equilibrium with the substrate. Following the work by Fredberg and colleagues, we performed these calculations under the assumption that the plate is homogeneously elastic (for details, see 'Assumptions for the thin-plate FEM approach' in 'Materials and methods' section) (*Tambe et al., 2011*, *2013*). Using the finite element method (FEM), we then determined the thin-plate internal stresses that are required to balance the traction stresses in the substrate. Once the thin-plate internal stresses are known, the local force exchange through any curvilinear section inside the plate can be calculated by multiplication of the stress tensor at a particular location on the section with the normal vector to the section line (*Figure 3A*). In our experiments, the curvilinear sections were defined by the fluorescence signals of GFP fused to E-cadherin, the molecular backbone of adherens junctions

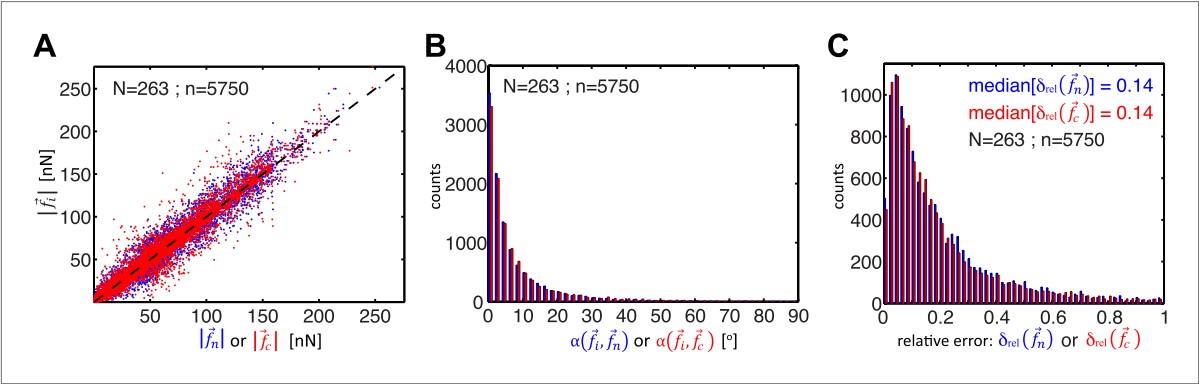

**Figure 2**. Comparison of cell–cell force calculations by force-balancing principle and by finite element modeling (FEM). (**A**) Comparison of cell–cell force magnitudes as measured by the force balancing method (blue, $\bar{f}_n$ is calculated from *Equation 5*) or FEM (red, $\bar{f}_c$) vs the two independent measurements at each interface as predicted by force balancing calculations ($\bar{f}_i = \bar{f}_{1,2}$ or $\bar{f}_{2,1}$). Deviations of individual data from the dashed line indicate measurement errors. (**B**) Angular deviations between cell–cell forces as measured by the force balancing method (blue, $\bar{f}_n$) or the FEM (red, $\bar{f}_c$) and the two independent measurements at each interface as predicted by the force balancing calculations ($\bar{f}_i$) (outliers (<3% of all data points) with angular deviations >90 degrees are not shown in the plot). (**C**) Relative error in cell–cell force measurements by the force balancing method (blue, $\bar{f}_n$) or the FEM method (red, $\bar{f}_c$). The median relative error for both measurement methods is 14% (outliers >100% relative error (<7% of all data points) are considered in the median calculation but are not shown in the plot). n = total number of measurements pooled from N distinct cell–cell junctions, with each junction having been measured over multiple time points.

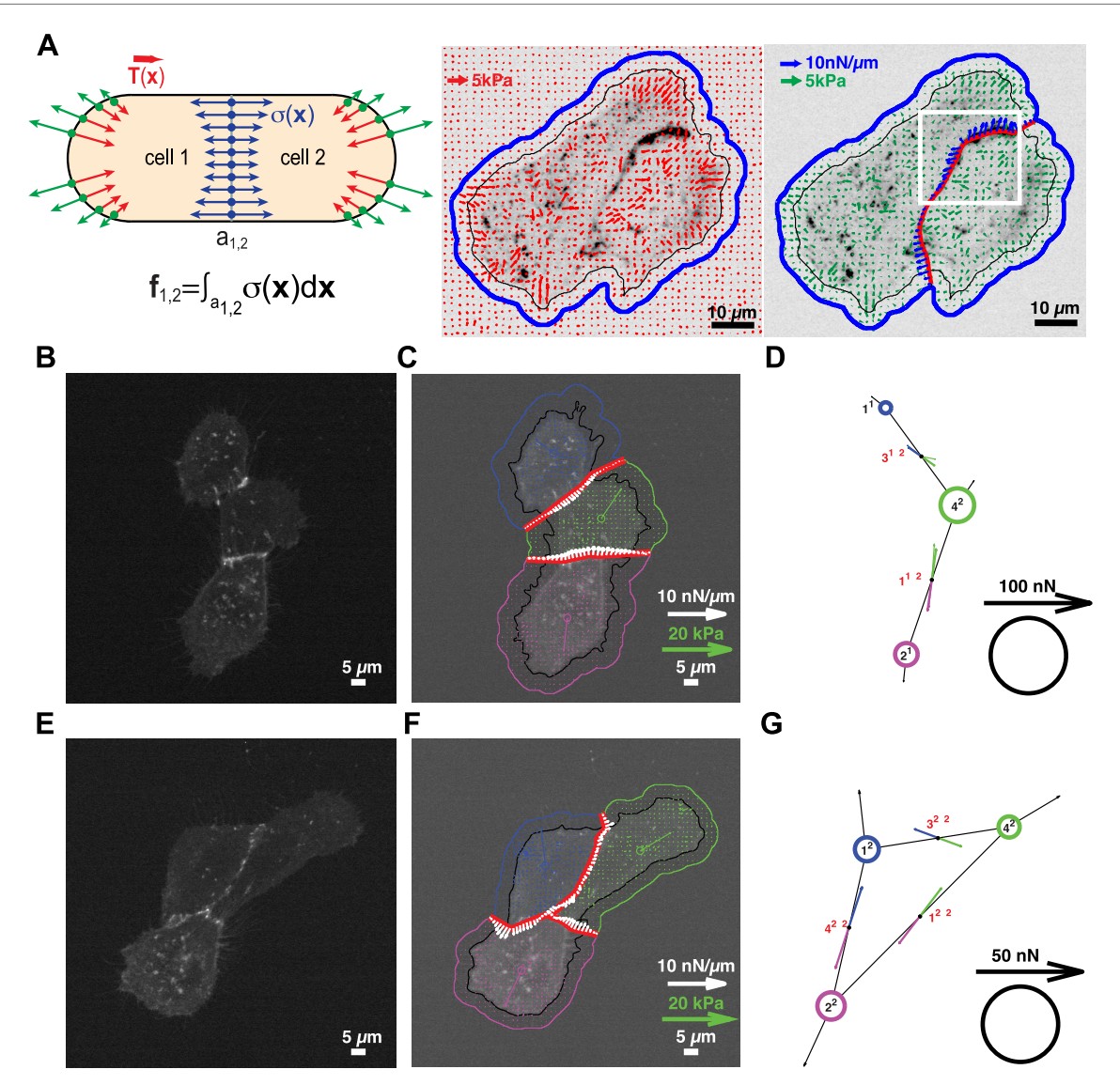

**Figure 3**. Quantification of cell–cell force exchange in a 3-cell cluster with changing topology. (**A**) Schematic of the thin-plate model implemented to determine the stress profile along a cell–cell junction. Sign-inverted traction forces (green vectors) are applied to the thin-plate model to generate an internal stress distribution of cells 1 and 2 that is consistent with the measured traction forces (red vectors). The internal stress distribution is then used to calculate the force profile along the cell–cell interface a$_{1,2}$ (blue vectors). See also **Equation 13** in 'Materials and methods' section. Images show application of the approach to a cell pair. The cell–cell interface is marked by E-cadherin-GFP (inverted fluorescence signal). See also **Video 1**. White box: region of interest highlighted in **Figure 6**. Regularized FTTC was used for traction force reconstruction. (**B**) Image of an E-cadherin-GFP-expressing 3-cell cluster with 'tree-like' topology, which permits the calculation of force exchanges at each cell–cell junction by both the force-balancing principle and the thin-plate FEM modeling. (**C**) Segmentation of cells in the cluster, overlaid on the traction force field (small colored vectors) and an inverted fluorescence image of the cell cluster. Longer vectors in cell centers indicate residual traction forces for individual cells. Cell–cell stresses (white arrows) were calculated from sign-inverted traction forces. (**D**) Graphical network representation of the cluster. Dashed arrows at graph edge midpoints indicate the cell–cell force vector obtained from the force-balancing principle. Solid arrows of the same color show the corresponding cell–cell force vector derived from the thin-plate model. The difference between the two force estimates indicates the combined uncertainty of the two methods. See **Figure 2** for error analysis over many clusters and junctions. (**E–G**) The same cluster as (**B–D**) at a different time point, when a junction formed between cell 1 and cell 2. This yields a loop topology preventing the calculation of cell–cell forces at junctions 1, 2, and 3 based on the force-balancing principle. See also **Video 2**.

The following figure supplement is available for figure 3:

**Figure supplement 1**. Quantification of cell–cell force exchange in a 6-cell cluster with changing topology.

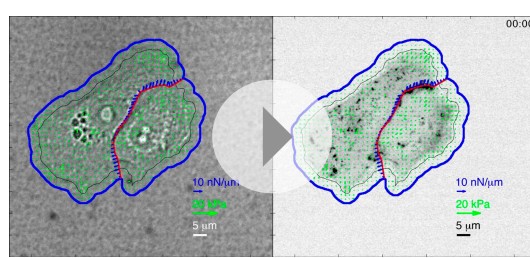

**Video 1.** Force exchange between E-cadherin-GFP-expressing MCF10A cell pair quantified at sub-junctional resolution using FEM; related to *Figure 3A*. Measured traction stresses (green vectors) were inverted for the calculation of mechanical stress distribution within the cell pair based on the thin-plate model, and the calculated cell internal stresses were integrated along the cell–cell junction to obtain the transmitted cell–cell stress profile along the junction (blue vectors; see 'Materials and methods' for details). Both the traction stress map and the stress profile were overlaid onto transmitted light (left) and E-cadherin-GFP fluorescence (right) images. The cell–cell junction is outlined by the red line, while the boundary of the cell pair is outlined by the black line (tight cluster mask). The blue outline represents the dilated cluster mask employed to capture the entire traction force field generated by the cell pair. Images were acquired one frame every 8 min for over 2 hr and 40 min with a 40× 0.95 NA air objective using a spinning disk confocal microscope; frame display rate =3 fps.

in epithelial tissues (*Halbleib and Nelson, 2006*). Thus we could calculate force exchange profiles along cell–cell interfaces (*Figure 3A*) at any time point of a time-lapse video (*Video 1*).

We tested the validity of the thin-plate model in cell clusters with a 'tree-like' topology, where the force exchange at cell–cell interfaces could be determined both by the force-balancing approach and integration of the FEM-predicted stress profile along the same interface (*Figure 1*). Our analyses indicated that both approaches yielded consistent results; the error for interface force calculations by the FEM approach is comparable to the error we found for force calculations by the force balancing principle alone (*Figure 2*). This implies that the inaccuracy in FEM-predicted forces arises from the same sources of error as the inaccuracy in force balance-predicted forces, which are uncertainties in the traction force reconstruction. Thus, neither the simplifying approximation of the cell cluster by a homogeneous thin-plate nor the numerical solution of the model introduced substantial error when predicting cell–cell forces in 'tree-like' topologies. Although both the force-balancing-principle approach and the FEM approach appeared to have similar accuracy in determining cell–cell forces, application of the thin-plate model permits the analysis of force exchange in clusters that change between a 'tree-like' and a 'loop' topology (*Figure 3C* and

*Figure 3—figure supplement 1B* vs *Figure 3B* and *Figure 3—figure supplement 1A*; *Videos 2 and 3*) and is thus a more generalized approach to quantify cell–cell force transmission.

## Quantification of cell–cell force dynamics in proliferating cell clusters

We first applied our generalized method to study the dynamics and patterns of cell–cell adhesion forces in naturally formed clusters of MCF10A epithelial cells in 2D adherent cultures, including clusters that were undergoing cell divisions (*Figure 4A–C*; *Video 4*). Cell proliferation is an important factor in regulating the tensional homeostasis of tissues (*Farhadifar et al., 2007*; *Ranft et al., 2010*; *Eisenhoffer et al., 2012*), and the process itself is finely coordinated by intracellular and extracellular forces (*Théry et al., 2007*; *Woolner and Papalopulu, 2012*). We found that, as a cell in a cluster entered mitosis, the overall mechanical energy deposited into the substrate by either the entire cluster or the dividing cell transiently decreased (*Figure 4D–E*). The overall mechanical energy generated by a cluster or individual cells therein was calculated as the strain energy, that is, the product of traction force and substrate deformation, integrated over the footprint of a cluster or a cell. The drop in strain energy during mitosis reflected the detachment of the dividing cell from the cell–matrix as it became rounded (*Video 4*), consistent with previous assertions (*Terasima and Tolmach, 1963*; *Burton and Taylor, 1997*; *Tanimoto and Sano, 2012*). After mitosis, the strain energy was restored and, in this example, increased beyond pre-mitotic levels (*Figure 4D*), possibly reflecting the additional contractility that the new cell contributed to the cluster.

Each mitotic event correlated with a dramatic decrease in the forces, which the dividing cell exchanged with its neighbors (*Figure 4F–H*), suggesting that mitosis is accompanied by weakening of not only cell–matrix adhesions but also cell–cell adhesions. Interestingly, during mitotic events, the sum of cell–cell force magnitudes over all junctions in the cell cluster remained constant before increasing to a higher plateau value (*Figure 4I*). This suggested that the loss of cell–cell forces at the junctions of the dividing cell was balanced at other junctions in the cluster (*Figure 4—figure supplement 1*). For example, during division of cell 1 in *Figure 4A–B*, the two connecting graphical edges, edges 1 and 2, exhibited decreases in cell–cell forces (*Figure 4F*). In contrast, edge 5, which was not directly connected

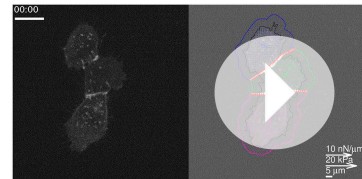
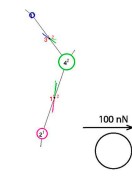

**Video 2**. Force exchanges in a three-cell MCF10A cluster with changing topology; related to **Figure 3**. (Left) E-cadherin-GFP fluorescence. (Middle) Cell–cell stresses (white vectors) and traction forces (colored vectors) overlaid on fluorescence images. The cell–cell junctions are outlined in red. Vectors originating from cell centers reflect the residual traction forces of each cell. See 'Materials and methods' for details. (Right) Network representation of the cluster indicating the residual traction force of each cell and the reconstructed cell–cell forces along junctions. Solid thick color vectors represent integrated cell–cell forces calculated using the FEM approach; dotted thin color vectors represent cell–cell forces calculated using the force-balancing principle. Note the force-balancing principle is insufficient for calculating junctions that are in a 'loop' configuration. Vector lengths and circle sizes represent force magnitudes. Images were acquired one frame every 4 min for 2 hr with a 40× 0.95 NA air objective using a spinning disk confocal microscope; frame display rate =3 fps.

to cell 1, experienced an increase in force transmission (**Figure 4G**; **Figure 4—figure supplement 1**). Similarly, during the division of cell 3 (**Figure 4B–C**), there was an increase in cell–cell force at edge 6 (**Figure 4F**; **Figure 4—figure supplement 1**), which was not connected to the dividing cell. These results show that, during major morphological changes, force fluctuations at individual cell–cell junctions can be dynamically compensated by forces at other cell junctions.

## Correlation of cell–cell force measurements with junctional localization of E-cadherin-GFP molecules

Our time-resolved force measurements demonstrated that cell–cell forces at individual junctions fluctuate over time, both during interphase and cell divisions. We thus investigated whether and how basal force variations were coupled to the localization and recruitment of the molecular components of cell–cell junctions. We focused first on E-cadherin, the backbone of adherens junctions for epithelial cells. We correlated the E-cadherin-GFP intensities integrated along each cell–cell junction with the force measured for each junction and observed a positive and statistically significant correlation (**Figure 5A**). The positive correlation also persisted on a stiffer matrix (**Figure 5A**), indicating that the overall relation between force exchange and E-cadherin recruitment is robust and independent of substrate compliance. We further validated the central role of cadherin proteins in mediating cell–cell force transmission by using function-blocking antibodies targeting E-cadherin directly (**Petrova et al., 2012**) or by knock-down of alpha-catenin, an essential adaptor protein of cadherin-mediated adhesions (**Yonemura et al., 2010**). Both perturbations resulted in a significant reduction in cell–cell force measurements (**Figure 5B–C**). These results support the model that the junctional recruitment of E-cadherin is coupled to the cell–cell adhesion forces (**Liu et al., 2010**; **Borghi et al., 2012**).

## Correlation of sub-junctional force exchange with local variations in E-cadherin-GFP recruitment

Visual inspection of the force profiles resolved by FEM and E-cadherin-GFP intensity distributions along the same cell–cell junctional interface unveiled a co-localization between high stresses with high intensity and lower stresses with low intensity (**Figure 6A**). This suggested that the coupling of force and E-cadherin recruitment observed at the level of entire cell–cell junctions could be translated to a sub-junctional length scale. Indeed, this visual impression could be confirmed quantitatively by a positive and statistically significant correlation between local E-cadherin-GFP intensities and cell–cell stress measurements (**Figure 6B**), both on soft and stiff substrates.

To validate our ability to measure cell–cell force exchanges with sub-cellular resolution, and to determine an approximate length scale over which force and E-cadherin recruitment are coupled, we divided each cell junction into shorter sub-junctional segments of different lengths and scrambled the intensity profiles within the segments. As the segment length increased, the randomization abrogated the positive correlation between cell–cell stress distribution and local E-cadherin-GFP intensities, as indicated by the decreasing ratio between correlation coefficients with and without randomization (**Figure 6C**, red curve). We defined the segment length at which the ratio between correlation coefficients with and without randomization fell below 0.5 as the length scale over which force exchange between cells and E-cadherin recruitment are coupled. We found this length to be 9.6 µm (**Figure 6C**) or 12.8 µm (**Figure 6—figure supplement 1A**) depending on whether the cells

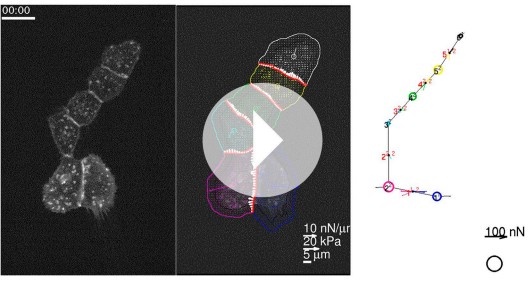

**Video 3**. Force exchanges in a six-cell MCF10A cluster with changing topology; related to *Figure 3—figure supplement 1*. (Left) E-cadherin-GFP fluorescence. (Middle) Cell–cell stresses (white vectors) and traction forces (colored vectors) overlaid on fluorescence images. The cell–cell junctions are outlined in red. Vectors originating from cell centers reflect the residual traction forces of each cell. See 'Materials and methods' for details. (Right) Network representation of the cluster indicating the residual traction force of each cell and the reconstructed cell–cell forces along junctions. Solid thick color vectors represent integrated cell–cell forces calculated using the FEM approach; dotted thin color vectors represent cell–cell forces calculated using the force-balancing principle. Note the force-balancing principle is insufficient for calculating junctions that are in a 'loop' configuration. Vector lengths and circle sizes represent force magnitudes. Images were acquired one frame every 7 min for >3 hr with a 40× 0.95 NA air objective using a spinning disk confocal microscope; frame display rate =3 fps.

were cultured on soft (8 kPa) or stiff (35 kPa) substrates. This length scale is consistent with the length scale over which E-cadherin-GFP proteins are modulated along cell–cell junctions in our system, which we measured to be 13 μm based on autocorrelation of E-cadherin-GFP intensities along the cell junctions (*Figure 6D*; *Figure 6—figure supplement 1B*). Similar values of cadherin intensity fluctuations along cell junctions have previously been reported for mature cell–cell adhesions (*Lambert et al., 2007*). Thus, our results suggest that the measured length scale of force–E-cadherin coupling is likely related to the spatial organization of E-cadherin along the cell–cell junctions and not a resolution limit of the FEM analysis. Importantly, the value ~10 μm is significantly less than the length of the majority of the cell–cell junctions in the MCF10A clusters (*Figure 6E*; *Figure 6—figure supplement 1C*). Even for junctions less than or equal to 10 μm, we were able to detect cell–cell stresses. In fact, the distribution of cell–cell stress magnitudes for these short junctions is similar to that for junctions of all lengths, indicating that our analysis of cell–cell stress is not limited by cell junction lengths (*Figure 6—figure supplement 2*). Together, these analyses demonstrate that the FEM approach is able to extract significant sub-junctional variations in cell–cell adhesion forces.

We also noted that for segments >40 μm, the correlation between local E-cadherin-GFP intensities and cell–cell stress measurements dramatically increased (*Figure 6C*, black curve). Since segments of this length could only be defined in extended, more mature cell–cell junctions (*Figure 6E*), we suspected that this trend could reflect a particular property of this more stable population of cell interfaces within the clusters. We therefore repeated the correlation analysis for bins containing junctions of increasing length. Indeed, we found that the median as well as the entire distribution of correlations between local cell–cell stress magnitudes and E-cadherin-GFP intensities increased with junction length (*Figure 6F*; *Figure 6—figure supplement 1D*). Importantly, although the number of junctions per bin decreased with length, the correlation increase is not related to weaker statistical power because randomization abrogated the trend (*Figure 6G*; *Figure 6—figure supplement 1E*). These results are consistent with more qualitative reports that mechanical forces exerted by actomyosin contractility are critical for junction growth and maturation (*Shewan et al., 2005*; *Yamada and Nelson, 2007*; *Borghi et al., 2012*; *Brieher and Yap, 2013*).

## Spatial patterns of cell–cell force distributions in cell clusters

Our measurements of cell–cell force fluctuations during mitosis also revealed spatial patterns of force transduction between multiple connected cells. When junctions were re-established after mitosis, those embedded in the cell cluster had lower force transmission compared to junctions at the cluster periphery (network in *Figure 4C*; *Figure 4—figure supplement 1B*). To further examine the spatial distribution of cell–cell forces, we categorized each cell in a cell cluster by the number of its neighbors or its degree of connectivity (k) (*Figure 1*). With increasing k, the sum of cell–cell forces increased (*Figure 7A*), which indicates that the cumulative force a cell experiences through its cell–cell adhesions increases with the number of connected neighbors. Strikingly, the increase in cumulative cell–cell forces for higher k-values was not paralleled by stronger cell–matrix traction forces (*Figure 7B*). Thus, contrary to the conclusions drawn from examining cell doublets (*Maruthamuthu et al., 2011*), the generation and exchange of forces at cell–cell junctions can be decoupled from cell–matrix

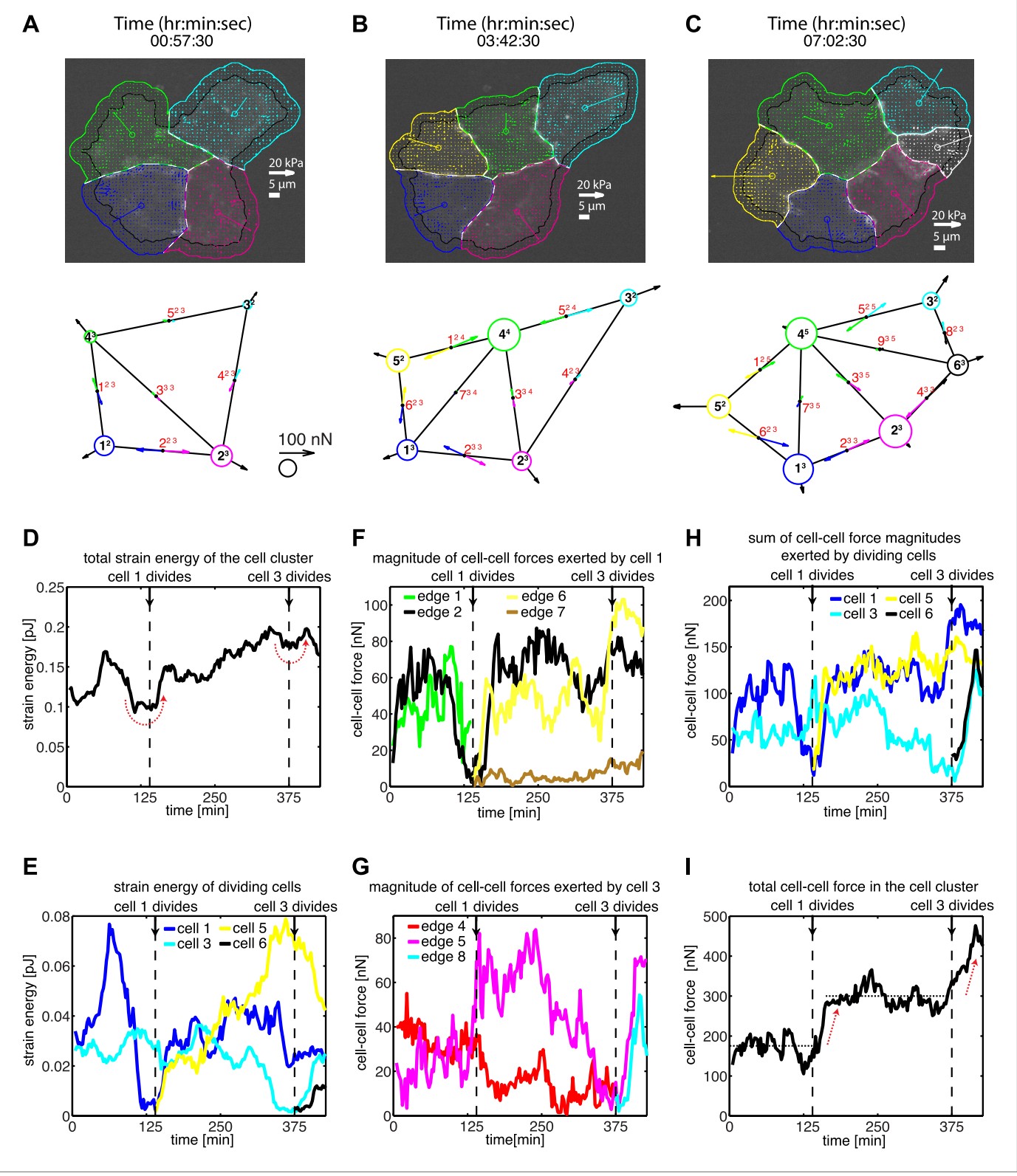

**Figure 4**. Force fluctuations in cell clusters during cell division. (**A–C**) A 4-cell cluster undergoing two cell divisions (cell 1 divides into 1 and 5; cell 3 divides into 3 and 6). Top row, fluorescent images of E-cadherin-GFP signals overlaid with traction force field. Vectors originating from the center of each cell reflect the residual traction force of the cell. Bottom row, graphical network representations including residual traction force for each cell and junctional cell–cell forces (see **Figure 1**). (**D–E**) Total strain energy on the substrate exerted by the cell cluster and cells 1 and 3 before, during, and after
*Figure 4. Continued on next page*

*Figure 4. Continued*

the cell division events. (**F**–**G**) Cell–cell force magnitudes exerted by cells 1 and 3 on each of their cell–cell junctions. (**H**–**I**) Sum of cell–cell force magnitudes exerted by the whole cluster and cells 1 and 3 before, during, and after the cell division events. See also *Figure 4—figure supplement 1* and *Video 4*.

The following figure supplement is available for figure 4:

**Figure supplement 1**. Force dynamics during cell divisions in cluster shown in *Figure 4*.

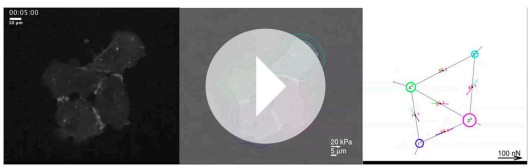

**Video 4**. Force exchange in a four-cell MCF10A cluster with two cell divisions; related to *Figure 4*. (Left) E-cadherin-GFP fluorescence. (Middle) E-cadherin-GFP fluorescence images overlaid with traction forces (small colored vectors). Tight cluster mask (black) and dilated (colors) cell outlines are also overlaid. Vectors originating from cell centers reflect the residual traction forces of each cell. (Right) Network representation of the cluster indicating the residual traction force of each cell and the reconstructed cell–cell forces along junctions. Vector lengths and circle sizes represent force magnitudes. Images were acquired one frame every 2.5 min over a time course of 7 hr with a 40× 0.95 NA air objective using a spinning disk confocal microscope; frame display rate =18 fps.

traction force generation, especially in larger cell clusters where cells have higher k values. This result is further supported by the observation that focal adhesions and traction forces are primarily localized at the periphery of cell clusters (*Figure 8*), consistent with a previous report (*Mertz et al., 2013*).

Although cell–matrix traction might not be directly coupled to cell–cell force transmission, it did nevertheless affect the distribution of cell–cell forces in cell clusters. We examined the forces transmitted through individual cell–cell junctions by assigning each junction, or graphical edge, the two k values of the two cells it connected. The cell–cell junctions were then grouped according to the lesser of the two k values (minimal k). Junctions with smaller minimal k values were closer to the cluster periphery, whereas those with higher minimal k were embedded in the cluster. In general, cells on stiffer (35 kPa) substrates exerted higher traction forces compared to cells on softer (8 kPa) substrates (*Figure 7B*). Interestingly, the dependence of traction force on matrix compliance was reflected in different trends of force transmission vs minimal k. On 8 kPa substrates, cell–cell forces at individual junctions were independent of the minimal k values (*Figure 7C*), whereas the cell–cell stresses, calculated by normalizing cell–cell force to junction length, increased for junctions with higher minimal k values (*Figure 7D*). This is consistent with a previous analysis of cellular stresses in epithelial sheets, in which the stresses exerted between cells were proposed to be higher towards the center of the sheet (*Trepat et al., 2009*). In contrast, on 35 kPa substrates, the forces as well as stresses exerted on individual cell–cell junctions decreased with increasing minimal k of the junction (*Figure 7C–D*). Hence, for cell clusters on the stiffer substrate, the cell–cell forces were transmitted predominantly between cluster-peripheral cells and not across the cluster center. Thus, the spatial pattern of cell–cell force transmission is modulated by the mechanical microenvironment.

## Analysis of force transmission through a cell

To obtain further insight into the spatial organization of cell–cell forces, we investigated the dynamic transmission of forces through cells. We performed pairwise cross-correlation analyses of forces between opposing cell–cell junctions and between forces at cell–cell junctions and cell–matrix adhesions (*Figure 9A*; see 'Materials and methods' section). This analysis is based on the following expectations of cell mechanics: if forces are transmitted across a cell without coupling to cell–matrix adhesions, the forces at the two considered junctions should fluctuate with similar magnitudes and opposite directions (correlation close to −1), whereas the fluctuations of cell–cell and cell–matrix forces should not be correlated (correlation close to 0). The opposite is true if forces exerted at one cell–cell junction are not transmitted across a cell but instead are transmitted to the cell substrate.

For control MCF10A cells, the coupling between cell–cell forces and cell–matrix was stronger than the coupling between cell–cell forces at opposing cell junctions, irrespective of substrate stiffness

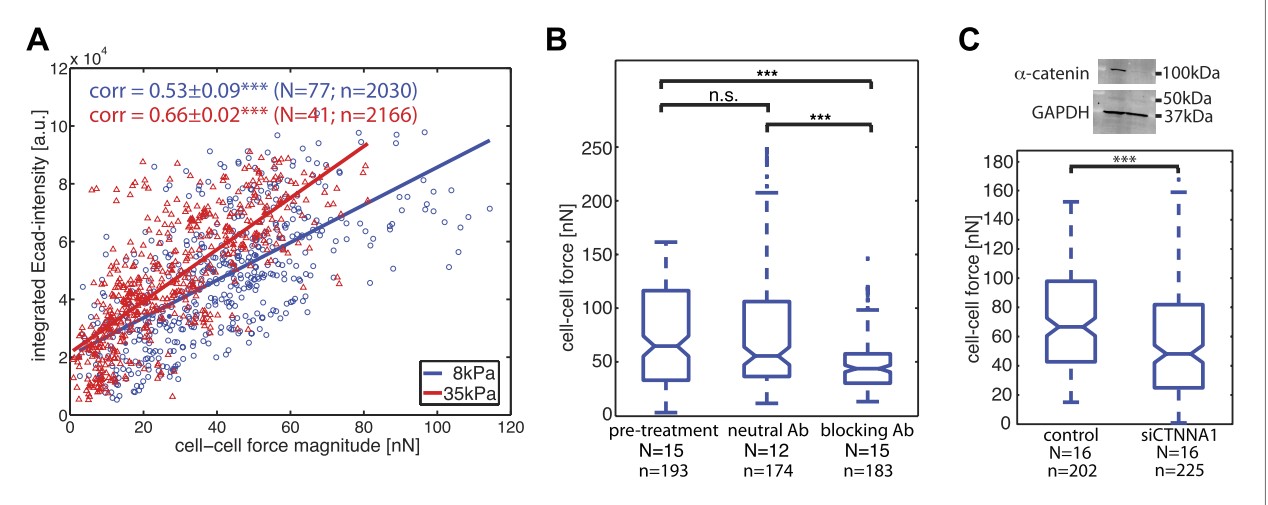

**Figure 5.** Correlation of calculated cell–cell forces with E-cadherin-GFP junctional localization. (**A**) Correlation between E-cadherin-GFP intensity integrated along cell–cell interfaces and the corresponding interfacial force (integrated force profile), for cell clusters cultured on 8 kPa and 35 kPa substrates. Correlation coefficients were calculated from n measurements from N distinct cell–cell junctions pooled from multiple independent experiments. Plot displays only a subset of n measurements from one experiment. (**B**) Forces between cells treated with neutral and blocking antibodies against E-cadherin compared to forces before treatment. (**C**) Top: Western blot showing downregulation of alpha-catenin in cells transfected with siCTNNA1. Bottom: cell–cell forces between cells transfected with siCTNNA1 compared to those between control cells. N = number of cell–cell junctions measured; n = total number of measurements from N junctions. ***p < 0.005.

(*Figure 9B*), indicating that forces exerted at cell–cell junctions were only weakly transmitted through a cell. At first, this result seemed to contradict findings that, in processes such as collective migration, mechanical interactions are long-ranged relative to the dimension of a single cell (*Trepat et al., 2009*). However, our data could be reconciled with this observation if long-range force transmission is an active process: forces exerted on a cell activate signaling pathways and contractile machineries that are responsible for transducing force to neighboring cells. In this model, each individual cell in a cell cluster, though linked with one another, functions independently with the ability to promote or attenuate force transduction.

There are two non-exclusive mechanisms that could attenuate force transduction across individual cells: first, forces at cell–cell junctions may be transmitted to the substrate via cell–matrix adhesions, thus intercepting the mechanical link between opposing cell–cell junctions ('force anchoring mechanism'). Second, each cell may have a basal actomyosin contractility level that is autonomous from extracellular force stimuli and high enough to overcome the cell-external forces ('force scrambling mechanism').

To examine these two possibilities, we measured cell–cell force transduction in mosaic cell clusters, where control cells were intermixed with cells in which paxillin, talin-1, or myosin-IIA were downregulated (*Figure 9*; *Figure 9—figure supplement 1*). Both paxillin and talin-1 are proteins involved in the assembly and maturation of integrin adhesions and the generation of traction forces (*Zaidel-Bar et al., 2004*; *Zhang et al., 2008*; *Iwanicki et al., 2011*). Therefore, we expected cells with downregulated paxillin or talin-1 to be mechanically isolated from the substrate. Indeed, the traction forces exerted by cells with reduced paxillin or talin-1 were significantly lower than the traction forces of control cells (*Figure 9—figure supplement 1D,H*). In cell pairs in which both cells had paxillin downregulated, forces at the cell–cell junction were unchanged compared to control cell pairs, indicating that paxillin knock-down did not affect cell–cell force transmission (*Figure 9—figure supplement 1C*). In talin-1-downregulated cell pairs, although cell–cell forces were lower than those of control cell pairs, cell–cell junction morphology was unaffected, and cell–cell force transmission was restored almost to control levels when the talin-1 downregulated cells were embedded in larger clusters (*Figure 9—figure supplement 1F,G,M*). Importantly, the correlation between the fluctuations of cell–cell forces in opposing junctions of a paxillin or talin-1 downregulated cell was higher than that between the cell–cell force fluctuations in opposing junctions of control cells (*Figure 9B*). For talin-1 downregulated cells, which exert very low traction forces (*Figure 9—figure supplement 1H*), the correlation between

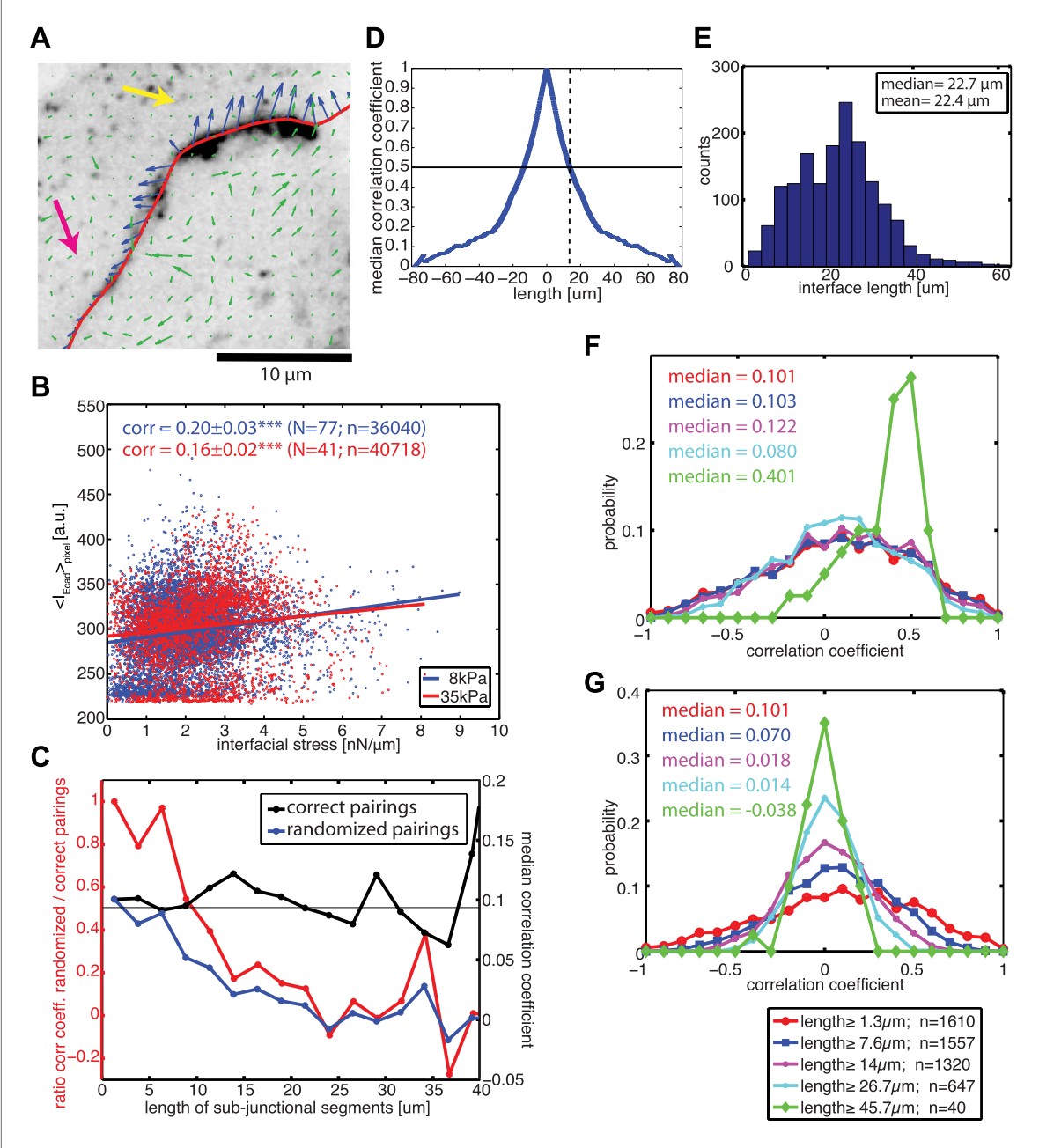

**Figure 6**. Correlation between calculated cell–cell forces at sub-junctional resolution and local E-cadherin-GFP intensities. (**A**) E-cadherin-GFP intensity along a cell–cell junction overlaid by cell–cell stresses (blue) calculated by FEM (magnified region of interest indicated in **Figure 3A**). Green vectors: traction forces; yellow and magenta arrows highlight sub-junctional segments where high and low E-cadherin-GFP intensity correlated with high and low forces, respectively. (**B**) Correlation between local E-cadherin-GFP intensities and local calculated cell–cell forces for cell clusters cultured on 8 kPa and 35 kPa substrates. Correlation coefficients were calculated from n measurements from N distinct cell–cell junctions pooled from 5 independent experiments. For visualization purposes, the plot displays only a subset of the measurements extracted from one experiment. (**C**) Median correlation coefficients for correlation between local E-cadherin-GFP intensities and cell–cell forces calculated from sub-junctional segments of various lengths (correct pairings). Local E-cadherin-GFP intensities were then randomized within the sub-junctional segments of various lengths and correlated with calculated cell–cell forces from the corresponding segments (randomized pairings). The length-scale over which cell–cell stresses and E-cadherin intensity are coupled is estimated as the minimal sub-junctional segment length for which the ratio between the median correlation coefficient of randomized pairings and the median correlation coefficient of correct pairings drops below 0.5 (gray horizontal line), that is, randomization in shorter sub-junctional segments has no effect. Results were calculated from 77 junctions of 14 cell clusters cultured on 8 kPa substrates. (**D**) Autocorrelation of E-cadherin-GFP intensities along the same 77 junctions. Dotted line shows the sub-junctional length where the median autocorrelation coefficient drops below 0.5 (horizontal line).
*Figure 6. Continued on next page*

*Figure 6. Continued*

(**E**) Distribution of cell–cell junction lengths of 77 junctions, each measured over multiple time points. (**F**) Distribution of correlation coefficients between local E-cadherin-GFP intensities and cell–cell forces calculated from junctions of different lengths. (**G**) Distribution of correlation coefficients between local E-cadherin-GFP intensities and cell–cell forces randomized within junctions of different lengths. n = total number of measurements from 77 junctions of 14 cell clusters cultured on 8 kPa substrates. Similar results were found for cell clusters cultured on 35 kPa substrates (***Figure 6—figure supplement 1***).

The following figure supplements are available for figure 6:

**Figure supplement 1**. Correlation between calculated cell–cell stresses and local E-cadherin-GFP intensities along junctions of cell clusters cultured on 35 kPa substrates.

**Figure supplement 2**. Relationship between cell–cell stress and junction length.

the fluctuations of cell–cell forces in opposing junctions was in addition higher than that between the cell–cell force and traction force fluctuations (***Figure 9B,D–E***; ***Video 5***). These data showed that the decoupling from the matrix of a cell within a cluster promotes the transmission of forces across the cell, from one cell–cell junction to the next, in support of the force anchoring mechanism as one way a cell may attenuate long-range force transmission.

The force scrambling mechanism was tested in mosaic cell clusters in which myosin-IIA, an isoform of the non-muscle myosin-II motor protein responsible for generating cell contractility (***Cai et al., 2010***), was downregulated in one or more cells. In cell pairs in which both cells had downregulated myosin-IIA, the traction forces exerted by the cells, as well as the forces exerted through the cell–cell junctions, were significantly decreased (***Figure 9—figure supplement 1K–L***). Although myosin-IIA downregulation affected cell–cell and cell–matrix mechanotransduction, cell–cell junctions were still able to form in cell clusters, as the cell–cell junction morphology of myosin-IIA downregulated cells remained similar to control cells and cells with downregulated myosin-IIA exerted forces at their cell–cell junctions close to control levels when embedded in larger clusters (***Figure 9—figure supplement 1J,N***). As with talin-1 downregulated cells, cells with myosin-IIA knock-down showed a significant increase in the correlation of force fluctuations at opposing junctions compared to control cells (***Figure 9B***). This shows that cells with reduced actomyosin contractility act as passive force transducers between other contractile cells in a cluster, consistent with the force scrambling mechanism.

## Discussion

Using our generalized cell–cell force measurement and analysis method, we quantified, spatially and temporally, the transmission of mechanical forces through cell–cell junctions of dynamic adherent epithelial clusters. Combined with image analyses of cell–cell junction configurations and rearrangements in the living cell clusters, our data provided unprecedented statistical power and time resolution to address questions regarding force distributions and fluctuations within cell clusters, as well as their coupling to the recruitment of molecular components of adherens junctions (***Maruthamuthu et al., 2011***).

To our knowledge, the proposed method is the first to resolve cell–cell force transmission dynamically with sub-cellular resolution. Yet, there are limitations that need to be considered with re-implementations of the approach: first, our method relies on a simple thin-plate model of the cell cluster that assumes homogeneity in material property within the cluster. In reality, the mechanical properties of the cell cluster may not be linear and isotropic elastic, as there may be spatial heterogeneity arising from differences in sub-cellular structures and cellular contractility. The assumption of mechanical homogeneity has been made also in stress distribution analyses on an entire cell monolayer (***Tambe et al., 2013***). In that case, the low spatial resolution tends to average inhomogeneity. The higher the resolution in imaging and the finer the spatial scale of the FEM analysis, the less likely is the cell cluster well represented by a thin-plate. How severe the resulting prediction error is depends on the cluster configuration and the noise level in the traction forces. In the present study, we validated the resolution of the inferred cell–cell forces by cross-correlation analysis of the spatial force fluctuations with E-cadherin density fluctuations along the junctions. These tests confirmed that despite the simple thin-plate model, cell–cell forces are resolved at the length scale E-cadherin-GFP intensities that fluctuate along the cell junction (***Figure 6***). We therefore conclude that the model simplifications are in this case not severe enough to obscure spatial relationships between force and underlying

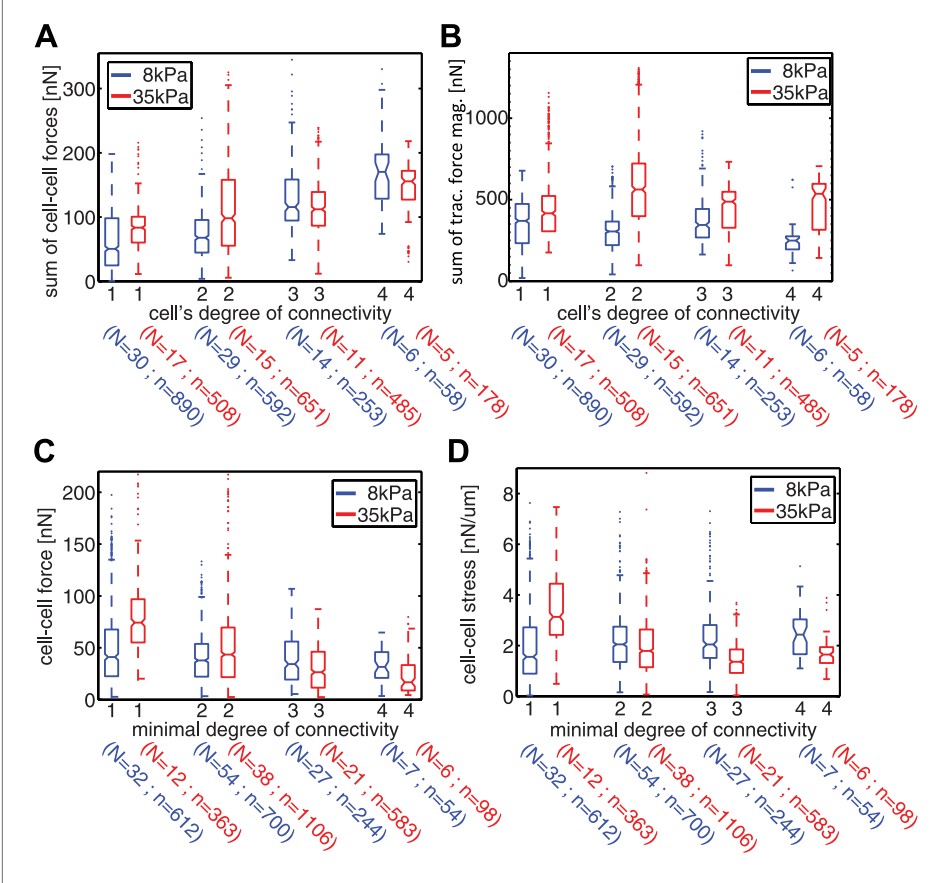

**Figure 7**. Spatial organization of cell–cell forces in clusters. (**A–B**) The sum of cell–cell force magnitudes (**A**) and traction force magnitudes (**B**) for cells connected to 1, 2, 3, or 4 neighbors (degree of connectivity k). (**C**) Force magnitudes at individual cell–cell junctions, classified according to the minimal degree of connectivity (smaller of the k values for the two connected cells). (**D**) Stress at individual cell–cell junctions, classified according to the minimal degree of connectivity. Center line within box represents median, notches indicate the 95% confidence interval about median. Non-overlapping notches between samples indicate that the sample medians differ with statistical significance at the 5% level. Lower and upper bounds of box indicate first and third quartiles. Whiskers indicate 1.5 times inter-quartile range. Points outside the whiskers represent outliers. n = total number of measurements from N distinct cells or junctions.

molecular processes in cell junctions. Second, our method relies on E-cadherin-GFP signals for determining the location of cell–cell junctions. Imaging resolutions (lateral and axial) as well as diffuse signals in dynamic junctions may affect the precision of determining the cell–cell boundary locations and therefore the accuracy of cell–cell force calculations. Third, the method relies on enclosure of the entire cluster in the field of view. Without tiling the image acquisition, the cluster size is therefore limited to a maximum of 8–10 cells. It is important to note that increasing cluster size would also increase the error of force measurement, as the error of traction force microscopy sums up during the calculation of cell–cell forces. Thus for detailed analyses of small relative force variations in space and time, it is advised to restrict the cluster size. We have quantified cell–cell force dynamics in naturally forming clusters of up to nine MCF10A cells. Lastly, because the method derives cell–cell forces indirectly through measurement of cell–matrix traction forces, a fundamental assumption is that the measured traction forces fully capture all forces produced by the cells. Thus, the method implicitly neglects the dissipation of forces inside cells, which—in principle—may affect cell–cell junctional forces. However, experiments that attempted to determine the magnitude of dissipative forces suggest that these contributions may be small compared to the elastic forces analyzed by the present approach (*Keren et al., 2009*).

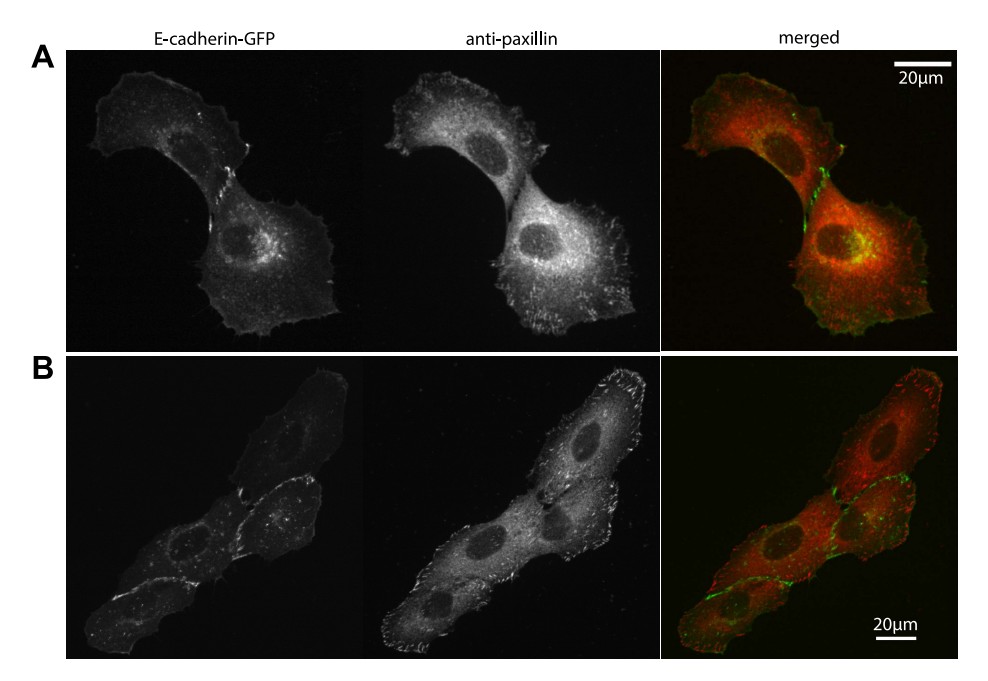

**Figure 8**. Localization of focal adhesions and cell–cell adhesions in MCF10A cell clusters. (**A**–**B**) Focal adhesions in a representative MCF10A cell pair (**A**) and 4-cell cluster visualized by immunostaining of paxillin, cell–cell junctions visualized by E-cadherin-GFP. Fluorescence images were acquired using the same parameters as for TFM measurements.

One key observation we made is that, at the multi-cellular level, junctional force distributions fluctuate temporally and spatially in response to local variations in cell–matrix adhesions and cellular contractility. Although tissue reorganization has long been known as mechanically regulated, our method provides now the capability to monitor the dynamics of cell–cell junctional force exchanges during various morphogenetic events and thus the ability to correlate the induction and outcome of processes such as cell polarization (*Blankenship et al., 2006*), mitotic spindle orientation (*Fink et al., 2011*), cell migration (*Toyama et al., 2008*), and cell sorting (*Landsberg et al., 2009*), not only with putative surrogates of force generation (e.g., myosin-II localization) but also with actual force values. We show that mechanical forces may be balanced during major topology-changing events such as cell division by dynamic redistribution of cell–cell junctional forces within small cell clusters (*Figure 4*). The result suggests that force fluctuations at individual cell–cell junctions can be dynamically compensated by forces at other cell junctions. We speculate that this type of mechanical compensation occurs during cell division—and apoptosis—within larger tissues as well and may be the basis of tissue stress homeostasis to maintain mechanical integrity in a proliferating and deforming epithelium.

Our method also provides insights into tissue mechanics by revealing that force transmission across a cell cluster is short-ranged, typically on the length scale of one cell diameter, due to the anchoring and scrambling of forces by cell–matrix adhesions and basal actomyosin contractility (*Figure 9*). The longer-range force transmission previously observed and required for tissue morphogenesis processes, such as collective cell migration (*Tambe et al., 2011*), is possible by the active regulation of force propagation. This can be accomplished by spatially coordinating the activation of actomyosin contractility above basal force levels, which yields a significant force exchange between neighboring cells, or by spatially coordinating the de-activation of actomyosin contractility and/or cell–matrix adhesion, which may increase the transduction range of basal force levels beyond one cell diameter. The notion of active propagation of mechanical forces across tissues via spatial gradients in myosin-II activation is supported by studies of epithelial sheet migration (*Ng et al., 2012*) and collective invasion of cancer cell clusters (*Hidalgo-Carcedo et al., 2011*), in which an asymmetric distribution of actomyosin contractility promote cell cohesion.

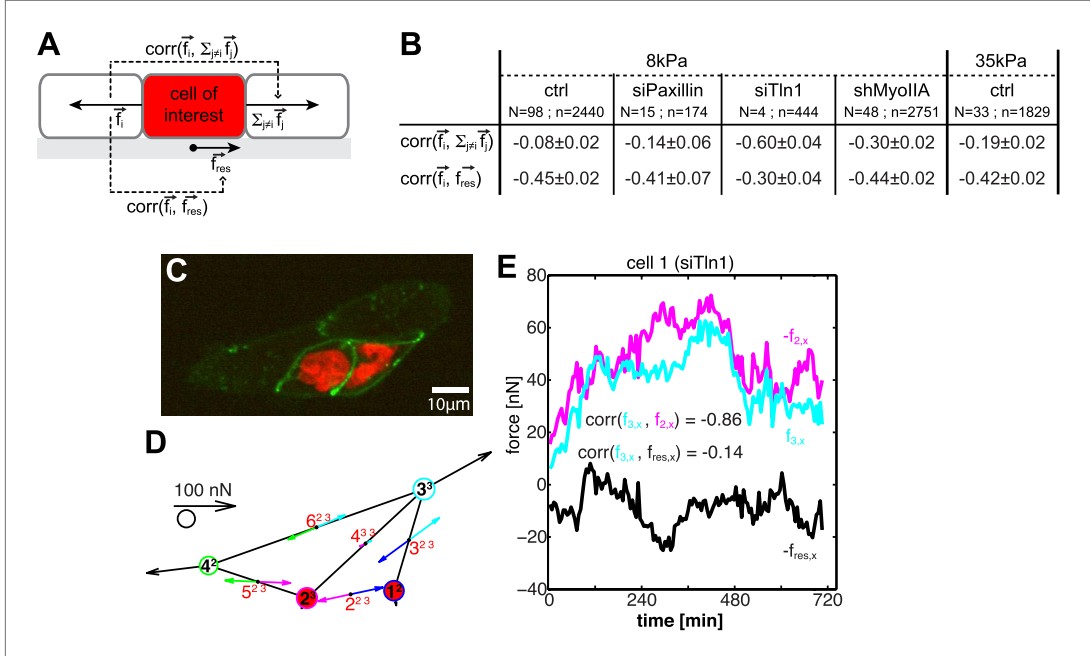

**Figure 9**. Dynamic force transmission through cells. (**A**) Schematic of temporal cross-correlation analysis of force fluctuations at opposing cell–cell junctions. To determine the extent of force transmission from one cell–cell junction across a cell (depicted here in red) to the next junction or to the cell substrate, force fluctuations at one cell–cell interface *i* of the cell-of-interest are correlated with the fluctuations of the vector sum of cell–cell forces at all remaining cell–cell junctions of this cell or with the fluctuations of the negative residual traction force of the cell, respectively. See 'Materials and methods' for details. (**B**) Cross-correlation analysis results for control cells on 8 kPa or 35 kPa substrates and for cells with downregulation of paxillin (siPax), talin-1 (siTln1), or myosin-IIA (shMyoIIA) in mosaic cell clusters on 8 kPa substrates. See *Figure 9—figure supplement 1*. (**C**) Mosaic cell cluster with two siTln1-treated cells (red nuclei). (**D**) Graphical network representation of the cluster at the same time point. See *Video 5* for full time lapse sequence. (**E**) Time courses of x-component of junctional forces (junction 2, magenta; junction 3, cyan) and residual traction force (black) in target cell 1 (cf. graphical network in **D**).

The following figure supplement is available for figure 9:

**Figure supplement 1**. Mosaic downregulation of paxillin, talin-1, and myosin-IIA.

Based on our statistical analysis of over 90 cell clusters and over 250 cell–cell junctions in these clusters, it appeared that intercellular force transmission exhibits distinct spatial patterns in clusters cultured on soft vs stiff substrates, in which cell–cell forces are transmitted along cluster peripheral cells on stiff substrates, but through the cluster center on soft substrates (*Figure 7*). We speculate that this may also be a result of the active interplay between cell–cell adhesions, cell–matrix adhesions, and basal actomyosin contractility to transmit, anchor, and scramble forces. As previously reported by the Dufresne group (*Mertz et al., 2012, 2013*) and shown here in *Figure 7* and *Figure 8*, focal adhesion numbers and traction force magnitudes are higher along the outer periphery of the cluster, suggesting that cell–cell forces are derived from cell–matrix forces that occur mainly at the cluster periphery. On stiffer substrates, where actomyosin contractility and traction force exertion are known to be high (*Discher et al., 2005*), the force anchoring and scrambling mechanisms are likely more active, preventing force transmission to the center of the cluster, whereas on softer substrates such mechanisms are less active, allowing force propagation through the cluster center. Although it is yet unclear how substrate stiffness precisely regulates cell–matrix adhesions, the cytoskeletal network, and traction force exertion in cell clusters, it has been shown that the differences induced by substrate stiffness can extend multiple cell layers into a cell sheet (*Ng et al., 2012*). Understanding how microenvironmental mechanical properties influence force transduction in epithelial cell groups may have implications not only for developmental processes but also for pathologies such as cancer where matrix stiffness is often altered (*Paszek et al., 2005*).

**Video 5**. Force exchange in four-cell cluster with mosaic downregulation of talin-1 on 8 kPa substrates; related to *Figure 9C–E*. (Left) E-cadherin-GFP fluorescence (green). Cells with talin-1 knock-down are also labeled with H2B-mCherry (red). (Middle) E-cadherin-GFP fluorescence images overlaid with traction forces (small colored vectors). Tight cluster mask (black) and dilated (colors) cell outlines are also overlaid. Vectors originating from cell centers reflect the residual traction forces of each cell. (Right) Network representation of the cluster indicating the residual traction force of each cell and the reconstructed cell–cell forces along junctions. Vector lengths and circle sizes represent force magnitudes. Images were acquired one frame every 4.5 min over a time course of >10 hr with a 40× 0.95 NA air objective using a spinning disk confocal microscope; frame display rate =18 fps.

Lastly, our cell–cell force microscopy method allowed us to correlate, in quantitative terms, cell–cell force transmission to intracellular molecular dynamics. We showed that, at the sub-cellular level, the variations in junctional forces can be coupled to turnover of E-cadherin-GFP at the junctions. Future studies can apply the same approach for analyses of other cell–cell junction molecules, including known force-responsive proteins, such as alpha-catenin (*Yonemura et al., 2010*), vinculin (*le Duc et al., 2010*; *Huveneers et al., 2012*; *Thomas et al., 2013*), and Eplin (*Abe and Takeichi, 2008*; *Taguchi et al., 2011*). With temporal and spatial resolutions at both the sub-cellular and multi-cellular levels, our method represents a tested toolkit for systematic investigation of the complex force homeostatic mechanisms required for the maintenance of a stress-resistant tissue.

## Materials and methods

### Cells

MCF10A expressing E-cadherin-GFP were generated by retroviral infection with LZBOB-neo-E-cadherin-GFP expression vector (gift of MJ Wheelock and AB Reynolds (*Fukumoto et al., 2008*)) and selected with neomyocin (300 µg/ml). For experiments with mosaic cell clusters containing control cells intermixed with cells with knock-down of myosin-IIA or talin-1, MCF10A-E-cadherin-GFP cells that also expressed H2B-mCherry were used, such that the nuclear marker could serve to distinguish between the control and the knock-down cells. MCF10A-E-cadherin-GFP cells that also expressed H2B-mCherry were generated by infection with the pBabe-H2B-mCherry retroviral vector and selected with hygromycin (300 µg/ml). All MCF10A cells were maintained in DMEM/F12 media supplemented with 5% horse serum, 20 ng/ml EGF, 0.5 mg/ml hydrocortisone, 100 ng/ml cholera toxin, 10 µg/ml insulin, and 1× penicillin/streptomycin, as described previously (http://brugge.med.harvard.edu/protocols) (*Debnath et al., 2003*).

### Antibodies

In experiments where antibodies were used to perturb E-cadherin-mediated cell–cell adhesions, MCF10A-E-cadherin-GFP cells were treated with 3 µg/ml of neutral, control antibody (76D5; gift of BM Gumbiner (*Petrova et al., 2012*)), or E-cadherin-blocking antibody (DECMA-1; Abcam) for at least 2 hr prior to TFM measurements. As an additional control, cells were also imaged and measured pre-treatment.

In immunostaining experiments to visualize focal adhesions in MCF10A-E-cadherin-GFP cell clusters, a purified mouse anti-paxillin antibody (BD Biosciences, San Jose, CA) was used as the primary antibody and the Alexa Fluor 568 goat anti-mouse antibody (Life Technologies, Grand Island, NY) was used as the secondary antibody.

### RNA interference (RNAi) of alpha-catenin, talin-1, and myosin-IIA

Downregulation of alpha-catenin in MCF10A-E-cadherin-GFP cells was achieved by transient transfection of the cells with a pool of four small interfering RNA (siRNA) duplexes targeting human CTNNA1 (M-010505-01; Dharmacon, Lafayette, CO). Downregulation of talin-1 in MCF10A-E-cadherin-GFP or MCF10A-E-cadherin-GFP-H2B-mCherry cells was achieved by transient transfection of the cells with a pool of four siRNA duplexes targeting human talin-1 (M-012949-00; Dharmacon). Transient transfections were performed as described previously (*Simpson et al., 2008*), with 25 nM of siRNA and the DharmaFECT 3 transfection reagent (Dharmacon). Experiments were conducted for 48–72 hr after transfection.

Downregulation of myosin-IIA in MCF10A-E-cadherin-GFP or MCF10A-E-cadherin-GFP-H2B-mCherry cells was achieved by lentiviral infection of a short-hairpin RNA (shRNA) plasmid containing non-muscle myosin IIA (Open Biosystems, Huntsville, AL). Two shRNA sequences were tested separately:

sequence 1: 5'-CCGG-GACAGCAATCTGTACCGCATT-CTCGAG-AATGCGGTACAGATTGCTGTC-TTTTT-3' (TRCN0000029466) and sequence 2: 5'-CCGG-CCGCGAAGTCAGCTCCCTAAA-CTCGAG-TTTAGGGAGCTGACTTCGCGG-TTTTT-3' (TRCN0000029467). Stably infected cells were selected with puromycin (2 µg/ml). Experiments were conducted 48–72 hr after transfection.

Knock-down of alpha-catenin, talin-1, and myosin-IIA was confirmed by immunoblotting per standard protocols and probed with antibodies against alpha-catenin (Sigma, St. Louis, MO), talin-1 (Cell Signaling, Danvers, MA), and myosin-IIA (Covance, Dedham, MA), respectively. At least two independent experiments were conducted for each condition.

## Preparation of polyacrylamide (PAA) gel substrates for traction force microscopy (TFM)

Fibronectin-coated PAA gels containing 0.2 µm fluorescent microspheres (Life Technologies) were prepared on glass-bottomed dishes, as described previously (*Pelham and Wang, 1997*). In brief, the glass surfaces were incubated with 0.1 N NaOH and air-dried. The surfaces were then incubated with 3-aminopropyltrimethoxysilane (Sigma) and 0.5% glutaraldehyde (Sigma), and washed with distilled $H_2O$ in between incubations. After drying, a drop of acrylamide/bis-acrylamide solution containing ammonium persulfate (BioRad, Hercules, CA), tetramethylethylenediamine (TEMED; Sigma), and 0.2 µm fluorescent microspheres was pipetted onto the modified glass surface. A coverslip was placed over the droplets to ensure a flat gel surface after polymerization. 10 µg/ml of fibronectin was coupled to the PAA substrates via the bi-functional crosslinker sulfosuccinimidyl hexanoate (sulfo-SANPAH; Pierce Biotechnology, Rockford, IL). Gels with elastic moduli of 8 kPa and 35 kPa were generated with acrylamide/bis-acrylamide ratios of 5%/0.3% and 10%/0.2%, respectively. The elastic moduli were measured using a rheometer (AR-G2; TA Instruments, New Castle, DE). To accommodate fluorescent imaging with GFP and mCherry markers in cells, dark red fluorescent beads (660/680; Life Technologies) were used, at a final concentration of 0.0032% by volume.

## Traction force microscopy and calculation of traction forces

Cells on PAA substrates were imaged with a multispectral multimode spinning disk confocal microscope consisting of a Nikon Ti-E inverted motorized microscope equipped with a custom built 37°C microscope incubator enclosure with 5% CO2 delivery, an integrated Perfect Focus System, a 40× 0.95NA Plan Apo objective, a Yokogawa CSU-X1 spinning disk confocal head with internal motorized high speed emission filter wheel and Spectral Applied Research Borealis modification for increased light throughput and illumination homogeneity, and a Hamamatsu ORCA-AG cooled CCD camera. GFP, mCherry, and dark red bead fluorescence were excited with the 491 nm, 561 nm, and 642 nm lasers, respectively, and collected with a quad 405/488/561/647 dichroic mirror (Semrock, Rochester, NY) and a 525/50, 620/60, or 700/75 emission filter (Chroma, Bellows Falls, VT). Images were acquired at the focal plane where the top-most layer of fluorescence beads was in focus, which also corresponded with the bottom-most part of the cells as visualized by E-cadherin-GFP. Images were acquired with MetaMorph software (MDS Analytical Technologies, Sunnyvale, CA). Cells were trypsinized with 0.25% trypsin after imaging to obtain a reference frame of unstrained bead positions.

For calculation of traction forces, square image blocks with a template size of 15–25 pixels = 2.5–4.1 µm (where the smaller template sizes were applied on the stiff 35 kPa substrates) were centered on each reference bead position, identified as intensity maxima in the reference frame. Bead displacements were defined as the x–y shift maximizing the cross-correlation score of these image blocks in a corresponding region of the deformed bead image. To minimize false positive template matching, bead displacements with insignificant maxima in the cross-correlation score function were rejected (for details see: *Ji and Danuser, 2005*).

Traction forces were reconstructed from the bead displacements using an implementation of the regularized Fourier Transform Traction Cytometry (FTTC) method (*Butler et al., 2002*) provided by *Sabass et al. (2008)*. The method is based on the assumption that the PAA substrate can be regarded as an isotropic, linear elastic, infinite half-space. This allows a closed-form formulation of the relationship between traction forces and substrate deformation using the Boussinesq Green function (*Landau and Lifshitz, 1970*). In order to solve the ill-posed inversion problem, we applied zero-order Tikhonov regularization (*Schwarz et al., 2002*; *Sabass et al., 2008*). The regularization parameter $\lambda^2$ ranged from $5 \times 10^{-8}$ to $10^{-6}$. The smaller values were used for the data on 35 kPa.

## The force-balancing principle and its application to larger cell clusters

The force-balancing principle dictates that, in the absence of significant inertial and frictional forces from the cell microenvironment, the traction forces exerted by an adherent cell must be balanced by the cell substrate. Extending this argument beyond single cells, a cell cluster and its substrate must also be in mechanical equilibrium. The integration of the traction force (and also torque) over the footprint $\Omega_c$ of the cell cluster thus has to equal zero (*Figure 1*; *Equation 1*). The net traction force over the whole-cell cluster must also equal the sum of the traction forces integrated over the footprint of each individual cell. For example, in the case of a two-cell cluster with cells $i$ and $j$, the net traction force of the cell pair, which should be zero, will equal the sum of traction forces integrated over the footprints of each of the two cells ($\Omega_i$ and $\Omega_j$).

$$\int_{\Omega_c} \vec{T}(\vec{x})d\vec{x} = 0 = \int_{\Omega_i} \vec{T}(\vec{x})d\vec{x} + \int_{\Omega_j} \vec{T}(\vec{x})d\vec{x} \qquad (1)$$

Here, $\vec{T}$ denotes the traction stress exerted by the cells.

In general, cells within a cluster all exert traction forces on the cell substrate. Thus, the integrated traction force for each individual cell within the cluster is non-zero (i.e., $\int_{\Omega_i} \vec{T}(\vec{x})d\vec{x} \neq 0$ and $\int_{\Omega_j} \vec{T}(\vec{x})d\vec{x} \neq 0$ in *Equation 1*). To maintain a net traction force of zero for the whole cluster, the non-zero integrated traction force of an individual cell must be balanced by a counter force from the neighboring cell(s). For a cell $i$ in a cluster, this balancing counter force can be expressed as the negative of its integrated traction force, which we defined as its residual traction force $\vec{f}_{res_i}$, where $\vec{f}_{res_i} = -\int_{\Omega_i} \vec{T}(\vec{x})d\vec{x}$. Because the balancing counter force is transmitted by physical interactions between cells through cell–cell junction(s), $\vec{f}_{res_i}$ also represents the vectorial sum of all cell–cell forces exerted by cell $i$ on its neighbors. In the case of a two-cell cluster, based on *Equation 1*:

$$\vec{f}_{i,j} = \vec{f}_{res_i} = -\int_{\Omega_i} \vec{T}(\vec{x})d\vec{x} = \int_{\Omega_j} \vec{T}(\vec{x})d\vec{x} \qquad (2)$$

Here, $\vec{f}_{i,j}$ denotes the force exerted by cell $i$ on cell $j$ and $\vec{f}_{i,j} = -\vec{f}_{j,i}$. Hence, by applying the force-balancing principle, we can derive the forces transmitted between two cells based on traction force measurements of the cells within those clusters.

The force-balancing principle can be applied to all cell clusters with either a linear configuration or with a 'tree-like' configuration (*Figure 1*). In these types of cell clusters, an imaginary cut through any of the cell–cell junctions, or edges in the network representation, would divide the cluster into two disconnected cell groups. Therefore, at each of the edges, the cell–cell force transmitted can be obtained by summing up the traction forces over each cell group:

$$\int_{\Omega_c} \vec{T}(\vec{x})d\vec{x} = \sum_{i \in g_1} \int_{\Omega_i} \vec{T}(\vec{x})d\vec{x} + \sum_{j \in g_2} \int_{\Omega_j} \vec{T}(\vec{x})d\vec{x} = 0 \qquad (3)$$

$$\vec{f}_{g_1,g_2} = -\sum_{i \in g_1} \int_{\Omega_i} \vec{T}(\vec{x})d\vec{x} = \sum_{j \in g_2} \int_{\Omega_j} \vec{T}(\vec{x})d\vec{x} \qquad (4)$$

Here, $\vec{f}_{g_1,g_2}$ denotes the cell–cell force that is transmitted through the interface which connects cell groups $g_1$ and $g_2$. In our measurements, we calculated $\vec{f}_{g_1,g_2}$ by taking the mean of the two independent summations of traction forces for the two cell groups:

$$\left\langle \vec{f}_{g_1,g_2} \right\rangle = \frac{1}{2}\left( \vec{f}_{g_1,g_2} - \vec{f}_{g_2,g_1} \right) = \frac{1}{2}\left( \sum_{i \in g_1} \int_{\Omega_i} \vec{T}(\vec{x})d\vec{x} - \sum_{j \in g_2} \int_{\Omega_j} \vec{T}(\vec{x})d\vec{x} \right) \qquad (5)$$

By reiterating the imaginary cut through all edges of the cell network, the force transmission at each cell–cell junction of a linear or 'tree-like' cell cluster can be determined. While previous studies either missed this generalized method or only applied it to a linear three-cell cluster (the simplest cell cluster with greater than two cells), our work has now demonstrated the applicability of this method to many complex cell clusters with three or more cells.

## Cell strain energy

To calculate the strain energy of cell $j$ in the cluster, we first calculated the displacement field $\bar{u}_j$ that is associated with the traction forces $\bar{T}$ generated by cell $j$ alone. This was achieved by the following integral over the footprint $\Omega_j$ of cell $j$:

$$\vec{u}_j(\vec{x}) = \int_{\Omega_j} \mathbf{G}(\vec{x} - \vec{x}') \cdot \vec{T}(\vec{x}')d\vec{x}' \tag{6}$$

Here, $\mathbf{G}$ is the Boussinesq Green function. The strain energy $U_j$ of cell $j$, which is a measure for cellular contractility (**Butler et al., 2002**), is then given by:

$$U_j = \frac{1}{2}\int_{\Omega_j} \vec{T}(\vec{x}) \cdot \vec{u}_j(\vec{x})d\vec{x} \tag{7}$$

Note that simply integrating over the measured displacement field $\bar{u}(\vec{x})$ instead of $\bar{u}_j(\vec{x})$ would yield incorrect results. This is due to the fact that elastic forces are long ranged. For a cell cluster, this means that forces from one cell will cause substrate deformations outside of its own footprint, for instance underneath the footprint of a neighboring cell. In other words, $\bar{u}(\vec{x})$ within $\Omega_j$ is not generated by cell $j$ alone but contains contributions from all cells in the cluster. The above method of calculating cell strain energy accounts for these mechanical subtleties and includes only the cell-own strains in the energy calculation. We do note that the method neglects the energy a cell might have to use to strain the substrate in the presence of other cells.

## Image segmentation of cell boundaries and cell–cell interface

The E-cadherin-GFP fluorescence intensity signal was used to segment the cluster boundary as well as the cluster internal cell–cell interfaces. The cell cluster boundary was segmented automatically by first smoothing the E-cadherin-GFP fluorescence intensity image with a Gaussian filter with a standard deviation of 5 pixels. The filtered image was then thresholded according to a 'first minimum after first maximum algorithm': the threshold value was selected by first finding the lowest intensity maximum in the image histogram, which largely corresponds to background pixels. The first minimum in the histogram after this maximum was selected as the threshold. The largest connected component of the binary image was identified as the cell cluster. To smoothen the cluster boundary and to fill small holes, we applied a closing operation using a disk with closure radius of 3 pixels as structuring element. Larger holes in the cluster that might arise during cell divisions or when cells join the cluster were not closed and were appropriately treated as cluster external space. The obtained *tight cluster mask* was then dilated by 15 to 45 pixels using a disk as structuring element, yielding the *dilated cluster mask*. The dilation was performed to ensure that all significant traction forces are included in the cell–cell force calculations. There are several reasons why significant traction forces may fall outside the segmented cluster region. First and foremost, the traction force reconstruction is limited by the spatial resolution of the bead tracking (~2.5–5 µm, dependent on the signal-to-noise ratio of the bead images and the substrate stiffness, with higher resolution for stiffer substrates). Second, cells at the cluster periphery often show finger-like and force-generating protrusions that are not detected by automatic cluster segmentation as they have a low E-cadherin-GFP concentration. Similarly, dividing cells exhibit retraction fibers that are barely detectable in the E-cadherin-GFP channel but seem to exert forces distant to the main cell body. The presence of these traction force vectors outside the cell footprint presents a dilemma: one would ideally want to cut off the thin-plate model at the tight cluster footprint, but then one would lose these 'external' forces when setting up the force balance for the cluster. We thus have taken an approach where we include these forces by extending the tight cluster mask but penalize forces with increasing distance to the detected cluster edge.

Cell–cell interfaces within cell clusters were drawn by hand to achieve highest reliability of the interface location. The hand drawing was guided by the E-cadherin-GFP intensity maximum along the interface curves. In regions with broader intensity curves, we additionally used the traction force maps to optimize the interface locations by avoiding traction force vectors pointing towards the interface. This criterion was deduced from the commonly accepted notion that cells do not push but only pull on the substrate. The interface drawings were also performed by three different people to avoid personal biases. The obtained cell footprints and boundaries were stored as pixelated information. The cell

position was defined as the center of mass of the cell footprint. The interface length was defined as the cumulative length of line segments that connect every tenth pixel of the interface curve. If the interface curve crossed a segmentation hole in the cluster, the pixels in the holes were not counted to avoid an overestimation of the interface length.

## Finite element method for cell–cell force measurements

The mechanical stress distribution within a cell cluster was inferred by modeling the ventral cell cortex as a deformable thin-plate, whose internal stress field $\sigma(\bar{x})$ balances the sign-inverted traction forces $\bar{T}(\bar{x})$ according to *Landau and Lifshitz. (1970)*:

$$\frac{\partial \sigma_{ik}}{\partial x_k} - T_i(\bar{x}) = 0 \text{ in } \Omega_c \tag{8}$$

$$\sigma_{ik} n_k = 0 \text{ on } \partial \Omega_c \tag{9}$$

Here, $\Omega_c$ is the dilated cluster footprint that contains all significant traction forces and $\bar{n}$ denotes the outward normal to the dilated cluster boundary $\partial \Omega_c$. Indices run from 1 to 2 for the x and y in-plane dimensions, and the Einstein summation convention is used. Under the assumption that the thin-plate behaves as a linear elastic and isotropic medium, the plate-internal stress field is related to the plate-internal deformation (*Landau and Lifshitz, 1970*) by:

$$\sigma_{ik} = \frac{E(\bar{x})}{(1+v)} \left( \varepsilon_{ik} + \frac{1}{(1-2v)} \varepsilon_{ll} \delta_{ik} \right), \tag{10}$$

where $\varepsilon(\bar{x})$ is the strain tensor, $\delta_{ik}$ is the Kronecker's delta symbol, and $E(\bar{x})$ and v describe the spatially variable Young's modulus and the Poisson ratio, respectively. We set the Poisson ratio to 0.5, reflecting that cell cortices are largely incompressible (*Boal, 2002*).

The strain tensor relates to the field of displacements $\bar{u}(\bar{x})$ as:

$$\varepsilon_{ik} = \frac{1}{2} \left( \frac{\partial u_i}{\partial x_k} + \frac{\partial u_k}{\partial x_i} \right) \tag{11}$$

Together, *Equations 8–11* define a boundary value problem composed of partial differential equations (PDEs) that determine how the inverted traction force field propagates through the thin-plate, that is, how stresses inside the thin-plate modeled cell cluster cortex must be distributed to explain the measured tractions on the substrate and, at the same time, how these stresses must deform the thin-plate. The latter relation depends on the material properties of the cell cluster cortex, which are largely unknown. However, we found that the plate-internal stress field is practically invariant to the choice of the Young's modulus, consistent with previous studies (*Tambe et al., 2011*). The reason for this is primarily that the solution of *Equations 8–11* depends on the same material model to convert traction forces into plate deformation as is used to invert plate deformation back into plate-internal stress. Therefore in a first approximation, the terms describing the material properties are canceled out by the conversion of cluster-external tractions into cluster-internal stress. To test the validity of this approach, we compared the cell–cell junctional stresses derived from the PDE solution (see below) to the junctional forces calculated by the force-balancing principle in tree-like structures where the latter approach has a well-defined solution. Using a thin-plate model with a spatially homogeneous Young's modulus within the cluster, we obtained junctional stresses with a relative error that was comparable to the one of the force-balancing solution (the relative errors are 14% for both methods, see *Figure 2*). This result demonstrates that the error of both methods mainly originates from the uncertainty of the traction force measurement and that the FEM solution of the thin-plate model did not introduce additional numerical errors (e.g., due to numerical stress integration along discretized interfaces). For tree-like clusters, the two methods are thus interchangeable. This validity check on tree-like cluster, however, cannot be directly generalized to looped configurations. To this end, an alternative, yet missing, experimental method would be needed for cross-validating the thin-plate model. A more detailed cell model could yield a different stress distribution along the interfaces that may improve the correlation of cell–cell stresses with E-cadherin-GFP intensity as well as the prediction of cell–cell

forces in looped configurations. We maintain that the material model assumptions are cell-type specific as epithelial cells, especially on softer substrates, tend to have relatively homogeneous cortical structures with few pronounced stress fibers. Moreover, the invariance of the PDE solution to the material properties may also be abolished under perturbation of certain pathways.

In addition to setting Young's modulus constant within the cluster, we assumed that it decays exponentially with increasing distance $d$ from the tight cluster mask:

$$E = E_0 e^{-\frac{d}{\lambda}} \tag{12}$$

This assumption implies that pixels outside the tight cluster mask are less likely to belong to the cluster footprint as the distance increases. The length scale $\lambda$ was set to 10 pixels =1.6 µm.

Given a PDE solution predicting the plate-internal stresses, cell–cell junctional forces $\bar{f}_{m,n}$ transmitted through the interface $a_{m,n}$ between any two cells $m$ and $n$ can be calculated as:

$$\left(\bar{f}_{m,n}\right)_i = -\int_{a_{m,n}} \sigma_{ik} dl_k \tag{13}$$

Here, $\left(\bar{f}_{m,n}\right)_i$ denotes the $i^{th}$-component of the cell–cell force and $d\bar{l}$ defines a vector of length $dl$ located on the cell–cell interface and pointing normal to the interface toward cell $n$. Of note, the interface $a_{m,n}$ can represent the entire junction between two cells or only a short stretch of it. We used the second approach to calculate the dense force exchange profiles along cell–cell junctions.

The boundary value problem was solved numerically using the finite element method (FEM). Specifically, we implemented *Equations 8–11* in MATLAB using the 'Partial Differential Equation' toolbox. The triangular mesh was generated from the dilated cluster boundary curve (every 10$^{th}$ pixel was considered) using the function *initmesh* with the 'Jiggle'-option to improve the mesh quality. The mesh was then refined twice with the function *refinemesh* and further improved with a final application of the function *jigglemesh*. The resulting mesh was much denser than the traction force mesh to minimize interpolation errors. The boundary value problem was solved over the generated mesh with the *assempde* function. An additional component of the PDE solution implicated numerical stabilization, which is necessary because of the experimental error in determining the traction forces. The error in the traction force measurement is reflected by the inequality

$$\int_{\Omega_c} \bar{T}(\bar{x})d\bar{x} \neq 0 \tag{14}$$

that is, the traction forces integrated over the entire cluster are non-zero. This would imply that the cluster is subjected to an accelerating force resulting in translocation of the cluster. However, a finite solution to the boundary value problem exists only if the traction forces perfectly cancel one another over the cluster area. To correct the instability introduced by *Equation 14* we added a term $\kappa \bar{u}(\bar{x})$ with $\frac{\kappa}{E_0} = 10^{-7}$ to the left hand side of *Equation 8*. The resulting stress field remains numerically unchanged as long as the κ-value is chosen small enough, such that $\kappa \bar{u}(\bar{x})$ is negligible compared to the other physical terms in *Equation 8*.

## Assumptions for the thin-plate FEM approach

The FEM approach includes additional assumptions for calculating how mechanical stresses redistribute within a cell cluster. This was done by modeling the cell cluster as a thin, homogeneous, and elastic plate that is set under tension by the traction stress at the cell–substrate interface (*Figure 3A*). The internal cell-stress distributions were then calculated using the FEM such that the internal stresses were consistent with the measured cellular traction forces (see 'Finite element method for cell–cell force measurements'). In the final step, the calculated cluster internal stresses were integrated along each cell–cell interface to obtain transmitted junctional forces.

The assumptions of a thin, homogeneous, and elastic plate are well justified, as the studied MCF10A cells were relatively flat when adhered and spread on the polyacrylamide (PAA) gel substrates. The cells were typically only ~1 µm thick (except at the nucleus), while the lateral dimensions were ~20 µm. Cells in the small clusters also did not show any significant, micron-sized structural organization of actin (such as actin stress fibers) along which mechanical stresses could propagate on the compliant PAA

gels, supporting the homogeneous plate assumption. Furthermore, as we were only interested in the equilibrium distribution of mechanical stress in the cluster at certain time points, a purely elastic, rather than viscoelastic, description of the cells is sufficient.

This model is very similar to the one used in monolayer stress microscopy. The main difference is the fact that in our approach, the cluster boundary is completely defined within the field of view. Therefore in our approach, a complete force balance equation for the cluster can be built up, which is impossible in monolayer stress microscopy. (Note that in monolayer stress microscopy, it is unknown which stresses are transmitted at the margins of the field of view occupied with cells.)

## Tracking of cells and cell–cell interfaces

Cells were tracked by a nearest neighbor assignment, that is, by linking cells with minimal distance between their cell centers in consecutive frames (*Burkard, 1999*). In case of a cell division, only the closest daughter was linked. The second daughter cell was considered as a new cell starting a new time course.

Cell interfaces were tracked by minimizing the sum of three Euclidian distances, the distances between the two endpoints and their proximal correspondents, and the distance between the centers of mass. Pairs of interfaces that minimized this cost function among all possible pairings were linked.

## Adaption of frame rates for correlation analysis

For the correlation analysis, all data were adapted to a fixed frame rate of 1/240 s. When a dataset was acquired with a higher frame rate, the time series were averaged over bins of 240 s. All time series were individually normalized to a variance of 1 before grouping several time series together.

## Cross-correlation of E-cadherin-GFP signals with cell–cell stresses

Cell–cell stress values were calculated at every 10th pixel along the cell–cell interfaces using the stress tensor σ:

$$\sigma_{INTF,i} = \sigma_{ik}n_k ,$$ (15)

where $\sigma_{INTF,i}$ is the $i$th-component of the interface stress and $n_k$ is the $k$th-component of the vector normal to the interface. The average distance of two neighboring cell–cell stress points was 1.27 μm. The E-cadherin-GFP images were smoothed using a plain average filter with circular support of radius 10 pixels. For each cell–cell stress value, a corresponding E-cadherin-GFP intensity value was extracted from the discretized interface curve.

The correlation coefficient of E-cadherin-GFP intensity and stress magnitude were then calculated as:

$$corr(I(\vec{x}), \sigma_{INTF}(\vec{x})) = \frac{cov(I(\vec{x}), \sigma_{INTF}(\vec{x}))}{\sqrt{var(I)var(\sigma_{INTF})}}$$ (16)

To obtain a mean correlation coefficient, all stress and intensity pairs of all interfaces of all clusters and over all frames of an experiment were grouped together (*Figure 6B*).

The significant correlation coefficient obtained in *Figure 6B* suggests that FEM allows us to resolve meaningful interfacial stresses with sub-junctional resolution. To test this assertion and to determine the length scale over which force and E-cadherin recruitment are coupled, we calculated correlation coefficients for every interface in each frame individually, either with correct or randomized stress–intensity pairs. For randomization, the interface was subdivided into sub-junctional segments of length Δl. If a cell–cell junction was shorter than Δl, it was not considered in the analysis. If the interface was longer but not in multiples of Δl, the remaining fragment was complemented with neighboring data points to precisely match Δl. Randomization was then performed within each segment and one 'randomized' correlation coefficient was calculated for the entire interface. The correlation coefficients obtained from individual interfaces were then grouped together in a histogram (see *Figure 6F*). For uncorrelated data, one expects a median correlation of 0. Furthermore, the longer the segment length the more will the coefficient distribution tighten up with a peak at zero correlation. The length scale of cadherin-stress coupling was defined as the value of Δl for which the median correlation coefficient of the randomized pairs falls below 50% of the correct pairing (*Figure 6C*); this is to say that for shorter segment lengths, the randomized and correct pairings exhibit similar correlations. We cannot identify whether the abrogation of distinct correlation coefficients is related to the absence of finer force and

cadherin co-variations or because FEM cannot resolve finer stress variations. Thus, this limit also may define an upper bound for the resolution of sub-junctional forces.

## Cross-correlation of force fluctuations to assess force transmission across a cell

To precisely quantify long-range force transmission in cell clusters, we performed a cross-correlation analysis of force pairs from two interfaces of a cell. Consider a cell that is surrounded by at least two neighboring cells, denoted as '$m$' for the 'middle' cell (*Figure 9A*). At any time point, the vectorial sum over all cell–cell and cell–matrix forces has to be balanced. A fluctuation in direction or magnitude of one cell–cell force vector exerted on cell $m$, say $\vec{f}_{i,m}$, thus has to be counterbalanced by an opposite fluctuation of either the traction forces of cell $m$, $\vec{f}_{res_m} = -\int_{\Omega_m} \vec{T}(\bar{x}) d\bar{x}$, or by the cell–cell forces at the remaining interfaces of cell $m$, $\sum_{j \neq i} \vec{f}_{j,m}$. If the cell–cell forces are completely transmitted across cell $m$, then the cross-correlation between the cell–cell forces should be perfectly anti-correlated, that is, $corr\left(\vec{f}_{i,m}, \sum_{j \neq i} \vec{f}_{j,m}\right) = -1$. At the same time, no correlation between cell–cell force and traction forces should be found: $corr\left(\vec{f}_{i,m}, \vec{f}_{res_m}\right) = 0$. In the case where cell–cell forces are not transmitted across cell $m$, the forces should be decoupled, and their correlation should vanish $corr\left(\vec{f}_{i,m}, \sum_{j \neq i} \vec{f}_{j,m}\right) = 0$, while the cell–cell and cell traction forces should be anti-correlated: $corr\left(\vec{f}_{i,m}, \vec{f}_{res_m}\right) = -1$. This anti-correlation means that the cell–cell force is locally counter-balanced by the cell's traction forces. Implementation of this cross-correlation analysis is described in detail in the below section ('Cross-correlation of cellular force vectors').

## Cross-correlation of cellular force vectors

All possible component-wise cross-correlations of two vectors in two dimensions can be expressed in matrix form as:

$$corr\left(\vec{f}_1, \vec{f}_2\right) = \begin{bmatrix} corr\left(f_{1,x}, f_{2,x}\right) & corr\left(f_{1,x}, f_{2,y}\right) \\ corr\left(f_{1,y}, f_{2,x}\right) & corr\left(f_{1,y}, f_{2,y}\right) \end{bmatrix}. \tag{17}$$

We note that the matrix is symmetric. In our specific case, the vectors are two cellular forces $\vec{f}_1$ and $\vec{f}_2$ that are balanced by a third cellular force, $\vec{f}_3$. For example, when considering the correlation of the cell–cell forces, $corr\left(\vec{f}_{i,m}, \sum_{j \neq i} \vec{f}_{j,m}\right)$, exerted on cell $m$, the $\vec{f}_1$ corresponds to $\vec{f}_{i,m}$, $\vec{f}_2$ corresponds to $\sum_{j \neq i} \vec{f}_{j,m}$, and $\vec{f}_3$ corresponds to $\vec{f}_{res_m}$, subject to the force balance $\vec{f}_{i,m} + \sum_{j \neq i} \vec{f}_{j,m} + \vec{f}_{res_m} = 0$. Thus, it is expected that the off-diagonal components of the matrix in *Equation 17* equal zero on average, and that only the diagonal components yield a significant correlation. Furthermore, as the choice of the coordinate system is arbitrary, the correlation of the two diagonal components should yield, on average, identical values and thus:

$$corr\left(\vec{f}_1, \vec{f}_2\right) = c_{f_1, f_2} \begin{bmatrix} 1 & 0 \\ 0 & 1 \end{bmatrix}, \tag{18}$$

where $c_{f_1, f_2}$ is the average correlation of the x or y components, respectively. Numerically we calculated $c_{f_1, f_2}$ as the mean of the two diagonal components:

$$c_{f_1, f_2} = \frac{1}{2}\left(corr\left(f_{1,x}, f_{2,x}\right) + corr\left(f_{1,y}, f_{2,y}\right)\right). \tag{19}$$

We note that $-1 \leq c_{f_1, f_2} \leq 1$. Accordingly, temporal cross-correlations of forces are calculated as:

$$corr\left(f_1(t+dt), f_2(t)\right) = \frac{1}{2}\left(\sum_{k=x,y} corr\left(f_{1,k}(t+dt), f_{2,k}(t)\right)\right). \tag{20}$$

## Statistical analyses

Standard box plots were created using MATLAB with the horizontal line within each box indicating the sample median, the bottom and top of the box representing the first and third quartile, respectively, and the whiskers extending to the most extreme data points that fall within 1.5 times the difference between the first and third quartile. Points outside the whiskers represent outliers. Non-overlapping notches between samples indicate that the sample medians differ with statistical significance at the 5% level.

As indicated, additional statistical tests were conducted to calculate the level of statistical significance. The Wilcoxon rank sum test, also known as the Mann–Whitney U test, was used.

## General computational requirements

All computational parts of this project were performed in MATLAB. The solution of the thin-plate model required functions of the 'Partial Differential Equations' toolbox of MATLAB, which are listed above (see 'Finite element method for cell–cell force measurements'). Each calculation step is not particularly demanding with respect to computational hardware. For instance, no large memory is required and the entire analysis can thus be performed on normal computers. With respect to computation time, the most demanding step by far is the detection of bead displacements that are the input for the traction force calculations. It is an established method based on correlative template matching and takes on the order of several hours per video. In contrast, solving the thin-plate model is much faster and takes only several minutes. We note however that the actual computation time will be highly dependent on the individual hardware specifications and software implementation.

## MATLAB codes for analysis

We have included with the manuscript the core MATLAB scripts that were used to analyze our data, in order to encourage the community to adapt our cell-cell force measurement approach for future studies (*Source code 1*). Please understand that our raw codes, as they are currently written and run, is the natural product of a project that has developed, grown and organically evolved over several years as the ideas matured and more data were collected. As such, the code is in a state that is adaptable by those adept at deciphering MATLAB routines, but in no way meant to be comprehensive and directly executable by the general readership. The core MATLAB programs included are:

calcElEnergies.m and its dependent functions, for calculating strain energies of the cell cluster (see "Cell strain energy" in the Methods section),

cutOutForceFieldManyCells.m and its dependent functions, for segmentation of cell clusters and cell-cell junctions (see "Image segmentation of cell boundaries and cell-cell interface" in the Methods section),

and clusterAnalysis.m and its dependent functions, for tracking of cells and cell-cell junctions (see "Tracking of cells and cell-cell interfaces" in the Methods section), and for calculation of cell-cell forces and stresses using the force-balancing and thin-plate FEM approaches (see "Finite element method for cell-cell force measurements" in the Methods section).

The MATLAB toolboxes required for execution of the included programs are:

Image Processing Toolbox,
Partial Differential Equation Toolbox,
Symbolic Math Toolbox,
and Statistics Toolbox.

At the point of publication the software bundle does not include methods for traction force calculation. However, a user-friendly and mathematically advanced software package for this purpose is under review (revision) elsewhere. It will be released as soon as this manuscript is accepted. It will be downloadable from our website lccb.hms.harvard.edu and contain among several options the bead tracking and Fourier Transform Traction Cytometry (FTTC) methods used for the here described work. Please make sure to regularly visit this website for updates of our software packages.

All other approaches used for our analysis are detailed in the Methods section, such that those who wish to reproduce the results can do so.

We hope our methodology and the results we have reported in this article will foster further interests and stimulate new hypotheses and studies on cell biomechanics and mechanotransduction.

## Acknowledgements

We thank the Nikon Imaging Center at Harvard Medical School and its staff for the use of microscopes. We also thank Tasha Fagan for assistance in tracing cell–cell junctions. E-cadherin-GFP construct was a gift of MJ Wheelock and A Reynolds; neutral and blocking antibodies against E-cadherin were gifts of B Gumbiner. This research was supported by the Cell Migration Consortium (NIH GM064346 to JS Brugge and G Danuser), the Breast Cancer Research Foundation (to JS Brugge), the Lee Jeans Foundation through the Entertainment Industry Foundation (to JS Brugge), and NIH R01 GM071868 (to G Danuser). MR Ng was supported in part by the NIH Cell and Developmental Biology Training grant (GM07226). A Besser was supported by the Deutsche Forschungsgemeinschaft fellowship (BE4547/1-1).

## Additional information

### Funding

| Funder | Grant reference number | Author |
| --- | --- | --- |
| National Institutes of Health | Cell Migration Consortium GM064346 | Joan S Brugge, Gaudenz Danuser |
| Breast Cancer Research Foundation | | Joan S Brugge |
| Lee Jeans Foundation | | Joan S Brugge |
| National Institutes of Health | GM071868 | Gaudenz Danuser |
| National Institutes of Health | Cell and Developmental Biology Training Grant GM07226 | Mei Rosa Ng |
| Deutsche Forschungsgemeinschaft | BE4547/1-1 | Achim Besser |

The funders had no role in study design, data collection and interpretation, or the decision to submit the work for publication.

### Author contributions

MRN, AB, Conception and design, Analysis and interpretation of data, Drafting or revising the article; JSB, GD, Conception and design, Analysis and interpretation of data, Drafting or revising the article

## Additional files

### Supplementary file

• Source code 1. MATLAB analysis codes.

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
