## [Decision Letter]

Thank you for sending your work entitled “Mechanical stress homeostasis in active epithelial cell clusters by dynamic cell-cell force transduction” for consideration at *eLife*. Your article has been favorably evaluated by Tony Hunter (Senior Editor), a Reviewing editor, and 3 reviewers.

The Reviewing editor and the reviewers discussed their comments before we reached this decision, and the Reviewing Editor has assembled the following comments to help you prepare a revised submission.

The reviewers all agree that the new method presented is cutting edge, timely and important. The method uses the principle of force balance to estimate cell-cell forces from cell-matrix forces and combines it with a thin plate modeling approach to extract a spatially resolved map of cell-cell forces in epithelial monolayers. This allows 1) sub cellular measurements at a higher resolution than could previously be achieved, and 2) spatially-resolved dynamic measurements for small clusters of any topology. The method is first validated for “tree-like” clusters, in which case it can be compared to the already established force imbalance method. The method is then applied to the generic case of “loop” clusters, in which case this is the only method that up to now can give results. An interesting variant of this experiment is the use of “mosaic clusters”, where a molecular player of monolayer mechanics is depleted in a subset of cells within the cluster.

The general comment from all reviewers is that this is primarily a methodological paper. The biological applications presented do not provide real mechanistic insight and are proof-of-principle investigations showing that the method can be useful to address important biological questions. *eLife* is interested in publishing cutting-edge methods papers; therefore we would like to encourage the submission of a revised version. However, the reviewers found that for the method to be useful and of interest to a broad community, the presentation needs to be substantially revised and the assumptions and limitations of the method clearly discussed. We would also like to encourage you to make the code allowing the implementation of the method available to the public.

Specifically, two main points need to be addressed:

1) Presentation of the method and its limitations:

A key assumption is that the cell monolayer behaves like a thin elastic sheet with only two elastic moduli (linear and isotropic elasticity). In classical monolayer stress microscopy, this might be a better assumption because in this case large sheets of cells are observed without cellular resolution. Here however, the authors focus on small clusters of cells where cell contractility, stiffness, ECM adhesion, etc., vary from cell to cell, as it is also evident from the supplementary movies. In the Materials and Methods section, the authors suggest that this assumption might not be as severe as one might think; eventually the method reconstructs (cell-cell) forces from (cell-matrix) forces, thus the details of the material model might somehow drop out because they are used twice, once in the forward and once in the backward direction. However, this is a rather vague argument, could it be backed up by more explicit theoretical considerations?

Also, two of the experiments presented, cell division and mosaic clusters, definitely describe a very heterogeneous situation. In the movie showing a cell division event, it is clear that the dividing cell even partially detaches from the cluster (a large gap between the membranes can be observed). Thus there are intermittent new external boundaries, which are not taken into account in the analysis. Could the authors comment on this?

Another strong assumption is that there is no dissipation of forces by cortical elements, due to shear forces for example. Could the authors discuss this assumption?

Limitations: it is clearly the only method currently allowing one to estimate forces in four-cell cluster in “loop” configuration. But how reliable would the estimation be in a ten-cell or hundred-cell group?

One of the strongest advantages of the method presented is that it makes subcellular measurements possible. However, the resolution (10 um) is not very high. The authors suggest that this length scale could be a cellular feature and “not necessarily a resolution limit of the FEM analysis”. Would the method allow for measurements with a higher resolutions for epithelia with shorter junctions than these displayed by the MCF cells used here?

Reviewer #2 provided a detailed list of points that would make the presentation of the method more accessible. These points are listed in the minor comments pasted below.

2) Implications and limitations of the experiments presented:

The authors observe that intercellular forces are relaxed around mitotic cells although the total intercellular force in the cluster is kept constant, suggesting that the local relaxation is compensated by intercellular force increase between neighboring cells. This leads the authors to propose that such stress homeostasis might be the ”basis of mechanical integrity in a proliferating and deforming epithelium“. However, the conclusions are based on a single example with two dividing cells; the experiment does not appear strong enough to justify such strong conclusions. Could the authors either expand the experimental investigation of force homeostasis or tone down the conclusions? Could they also discuss what kind of mechanisms could lead to the proposed stress homeostasis?

One of the most striking findings of the paper is that intercellular forces are transmitted along cluster peripheral cell on a stiff substrate (on which traction forces are high), and through the center of the cluster on a soft substrate (on which traction forces are low). Why are cells and stress not shown here? Here again, could the authors discuss a potential mechanistic basis for this behavior?

It is not clear what the advantage of studying E-cadherin dynamics in such large clusters is. The method provides a relatively rough estimate of the force (14% precision as documented by the authors in a four-cell case, but probably less in larger clusters) whereas the correlation between force and E-cadherin concentration is a highly debated issue that requires exact force measurement. To be fully convincing, the positive correlation between force and E-cadherin and the increased correlation in growing and shrinking junctions, should be confirmed in cell doublets or linear triplets with exact intercellular force measurements. The time correlations displayed in Figure 7 are particularly inconclusive. Could the authors either substantiate these findings with more convincing experiments or remove these experiments? It does not seem correct to discuss a trend that is not substantiated by statistics (particularly compared to the statistics of the other experiments presented in the paper).

Furthermore, the movies show strong protrusive activity, with contacts between cells in the plane of imaging, in contrast with the 'standard' geometry of adherens junctions. Yet, the determination of cell-cell transmission uses in plane traction force measurements and assumes traction forces perpendicular to the interface.

The observed correlation between changes in E-cadherin intensity and stresses might reflect this stronger protrusive activity. How do the authors cope with this?

Minor comments:

The title does not seem to reflect the message of the paper; it suggests that the paper is about stress homeostasis, which is only one of the applications presented and which is not investigated in detail. We would suggest modifying the title to make clear that this is primarily a methods paper.

Could the authors provide details on how the force extraction is exactly done? What is the code used? What are the required inputs? Can the calculation be performed with a normal computer and how long does it take?

Reviewer #1:

The magnitude of the total intercellular force per cell depends on the numbers of neighboring cells, whereas the magnitude of the traction force on the ECM does not. Why are cells and stress not shown here? How reliable are cell-cell force measurements by FEM in clusters containing more than 20 cells?

Reviewer #2:

This manuscript is written in a very technical manner and the methods part at the beginning of the Results section but also later parts of the Results section are not easy to follow. Figure 1 is over-crowded, it is not clear to me why the mathematical equations and the references are required here. In order to demonstrate the way the method works, it seems more helpful to proceed to Figure 3 and to place Figure 2 later or in the supplement. The quantities analyzed in Figure 2 are not defined well and it is not clear how the numbers for N and n correspond to each other. In Figure 3 it should be explained why the sign inverted traction forces are shown. I also note that the network representation is rather dense and maybe even redundant, because the connectivity information on the edges can also be read off from the nodes (I understand that this information is used later to analyze force distribution and I do not mind leaving it in if the authors find this essential, but it shows again that the presentation is very crowded). What seems to be missing from this representation, though, seems to be information if a given node has resulted from a cell division. In Figure 4, the authors use a very special definition of strain energy (using only the displacement resulting from a single cell) without discussing alternatives or limitations. Regarding the length scale analysed in Figure 6, I wonder if the authors effectively extract the persistence length of the cell-cell contour; the explanations of the meaning of these length scales are rather hard to follow. In general, the authors should rethink how to make the presentation of their results more accessible to the general reader. I also note that Title and Abstract might be further sharpened. The used of the word “active” is not clear on first reading.

Although some details of the model are explained in large detail, the core of the method, namely the FEM-work, is not described in sufficient detail in the Materials and methods. Is the method implemented in the MATLAB PDE toolbox an optimization technique and if so, with which target function? What is the difference to the approach in monolayer stress microscopy, where one integrates the partial derivatives? Another issue is the use of an exponentially decaying Young's modulus between tight cluster mask and footprint boundary. Why is this complicated ad hoc procedure necessary? Can it happen that the Young's modulus has already decayed to a very small value at the cero stress boundary of the footprint and would this not lead to artifacts?

Regarding the cell-matrix force reconstruction, the authors use standard approaches and there is no real concern here, except that the method used (regularized FTTC) should be mentioned in the main text (at least when showing the traction maps in Figure 3, which obviously use a cubic lattice) and that a number should be given for the spatial resolution (actually the bead density seems to be quite low). What is the maximal bead displacement used in these experiments? How could the spatial resolution be improved in the future (leading to a smaller footprint region and to better results regarding the adhesions)? Such a discussion is important to understand the limitations of this approach and would also be helpful to understand the complicated procedure used to extract the sub-cellular length scale.

Why do global patterns of force distribution arise when at the same time the authors argue that force transmission is locally scrambled by cell-matrix contacts and actomyosin contractility? In fact these results seem to be contradictory, because stress localization to the rim is exactly what one expects from thin plate theory with long-ranged forces; because the authors use this assumption in their method, it is not clear how consistent their conclusions are in regard to the method used.

---

## [Author Response]

*1) Presentation of the method and its limitations*:

*A key assumption is that the cell monolayer behaves like a thin elastic sheet with only two elastic moduli (linear and isotropic elasticity). In classical monolayer stress microscopy, this might be a better assumption because in this case large sheets of cells are observed without cellular resolution. Here however, the authors focus on small clusters of cells where cell contractility, stiffness, ECM adhesion, etc., vary from cell to cell, as it is also evident from the supplementary movies. In the Materials and methods section, the authors suggest that this assumption might not be as severe as one might think; eventually the method reconstructs (cell-cell) forces from (cell-matrix) forces, thus the details of the material model might somehow drop out because they are used twice, once in the forward and once in the backward direction. However, this is a rather vague argument, could it be backed up by more explicit theoretical considerations*?

We have now included in the Discussion section a discussion of the limitations of the method. We note that, as the reviewers pointed out, the assumption of the cell cluster as a thin elastic sheet is an approach used in monolayer stress microscopy by Fredberg and colleagues, who have explicitly described how the material model of the thin plate representing the cell sheet is in fact irrelevant for reconstructing cell-cell stress distribution in the cell sheet (Tambe et al., Nature Materials, 2011). We agree with the reviewers that it will be interesting to test how a material model that does not assume homogeneity in elastic moduli within the cell cluster would alter the cell-cell force measurements; however, at this point, i.e. in the absence of independent measurements of the distribution of the visco-elastic tensor, it is rather unclear how to go about formulating such a refined model. At best, we could visualize cytoskeletal and focal adhesion proteins in the dynamic cell clusters. But it is still an enormous leap to define how the distribution of these proteins correlates with inhomogeneity in material property in a cell cluster. The critical point of this study is that we can show that despite the simplicity of the thin plate model, it is still possible to resolve cell-cell forces at a resolution that is biologically relevant, i.e. at the length scale of E-cadherin self-correlation at cell-cell junctions (Figure 6). This suggests that the inhomogeneity in the material properties, even for the analysis of subcellular force distributions, plays only a secondary role.

*Also, two of the experiments presented, cell division and mosaic clusters, definitely describe a very heterogeneous situation. In the movie showing a cell division event, it is clear that the dividing cell even partially detaches from the cluster (a large gap between the membranes can be observed). Thus there are intermittent new external boundaries, which are not taken into account in the analysis. Could the authors comment on this*?

The ability to capture and measure cell-cell forces during these dynamic events is a strength of our method: we used semi-automatic image segmentation to detect the boundaries of the cell clusters, in which the cell-cell boundaries were visually validated and manually corrected based on E-cadherin-GFP signals, if necessary. If there are gaps between the membranes, the algorithm will present the gaps as “holes” in the cell cluster, and segments of “cell junctions” within the “holes” are treated the very same way as junctions that are outside the tight cluster mask (see supplement for definition). Usually the gaps are so small, that they are still part of the dilated cluster mask, and are thus included in the cell plate model, albeit with a modulus that decays exponentially with Euclidian distance to the tight cluster mask. The algorithm may thus associate cell-cell stress with junction segments within such gaps. Note that for calculating junction lengths, these segments are not taken into account, which is again consistent with the treatment of other junction segments that are part of the dilated (but not of the tight) cluster mask. As these small gaps between the membranes are usually transient and the connection between cells maintained, it is reasonable not to fully neglect these regions during force calculations. We would like to note that, in the example shown in Figure 4 and Video 4, although there was temporarily a gap at junction 7 after the division of cell 1 into cells 1 and 5, our method correctly indicate that the force transmission at junction 7 is extremely low.

*Another strong assumption is that there is no dissipation of forces by cortical elements, due to shear forces for example. Could the authors discuss this assumption*?

In fact, our approach takes in-plane shear forces into account irrespectively of whether these forces are of elastic or viscous (dissipative) nature because the overall force balance with the substrate (and with the measured traction forces) must hold. For example, if two cells slide relative to each other and thereby exert frictional forces, then these forces have to be balanced by either cell through traction forces that define the measured input to our approach. The normal vs the shear component of forces at cell-cell junctions is fully encoded in the stress tensor.

*Limitations: it is clearly the only method currently allowing one to estimate forces in four-cell cluster in “loop” configuration. But how reliable would the estimation be in a ten-cell or hundred-cell group*?

We have now discussed this limitation in the Discussion section; the maximum cluster size for which we can measure cell-cell forces is limited by the field of view of the microscope, because our method requires visualization of the cell cluster boundary to minimize errors during force reconstruction. Increasing cluster size also increases the error of cell-cell force measurement as the error of traction force measurement adds up in the analysis. We have so far been able to image and measure cell-cell forces for MCF10A cell clusters of up to 9 cells.

*One of the strongest advantages of the method presented is that it makes subcellular measurements possible. However, the resolution (10 um) is not very high. The authors suggest that this length scale could be a cellular feature and “not necessarily a resolution limit of the FEM analysis”. Would the method allow for measurements with a higher resolutions for epithelia with shorter junctions than these displayed by the MCF cells used here*?

As we report in the paper, we conclude that the 10µm “resolution” is not necessarily a resolution limit of our method, but rather the smallest length scale at which we detect cell-cell force fluctuations along cell junctions of MCF10A cells, likely due to the length scale at which E-cadherin molecules are organized along the cell junction (measured by auto-correlation of E-cadherin-GFP fluorescence intensities along the junction). From this, it would follow that in a cell model with a finer granularity in junction organization, the method potentially could extract shorter scale force fluctuations. We note that, of course, this argument does not directly address the reviewer’s question. His/her question is one regarding the prediction of detectability (i.e. how short can a junction be to have a detectable cell-cell force) from the resolution. Usually, the detectability is significantly higher than the resolution (a principle used in optics to implement ‘super-resolution’ approaches like STORM or PALM). In our manuscript, we have measured cell-cell forces in many junctions shorter than 10 um (Figure 6 and Figure 6—figure supplement 1). To examine the cell-cell stress we were able to measure from these short junctions, we plotted below cell-cell stress magnitudes as a function of cell junction lengths for junctions with a minimal degree of 1 (such as cell pairs or linear clusters). Only junctions with a minimal degree of 1 were used to avoid potential false attribution of cell-cell stress from adjoining junctions. We noted that the distribution of cell-cell stress magnitudes for these junctions shorter or equal to 10µm is similar to that for junctions of all lengths, indicating that the analysis of cell-cell stress is independent of junction lengths.Author response image 1.

*2) Implications and limitations of the experiments presented*:

*The authors observe that intercellular forces are relaxed around mitotic cells although the total intercellular force in the cluster is kept constant, suggesting that the local relaxation is compensated by intercellular force increase between neighboring cells. This leads the authors to propose that such stress homeostasis might be the “basis of mechanical integrity in a proliferating and deforming epithelium”. However, the conclusions are based on a single example with two dividing cells; the experiment does not appear strong enough to justify such strong conclusions. Could the authors either expand the experimental investigation of force homeostasis or tone down the conclusions*?

We thank the reviewers for their suggestion and have revised the manuscript by clearly discussing the limitations of the method in the Discussion section, and by either removing any conclusions that may appear too strong or clearly indicating the speculative nature of statements such as the proposal of force re-distribution as a mechanism of stress homeostasis. We hope the reviewers now find the biological conclusions of our revised manuscript more modest, yet still thought-provoking. We were very excited by many of our biological findings enabled by our new method, and hope that the manuscript will provide the community with a new approach as well as some results for future hypotheses and investigations.

*Could they also discuss what kind of mechanisms could lead to the proposed stress homeostasis*?

As we discussed, the dynamic re-distribution/ compensation of cell-cell forces as a mechanism for “stress homeostasis” is still speculative. We will therefore refrain from further speculating on potential mechanisms.

*One of the most striking findings of the paper is that intercellular forces are transmitted along cluster peripheral cell on a stiff substrate (on which traction forces are high), and through the center of the cluster on a soft substrate (on which traction forces are low). Why are cells and stress not shown here? Here again, could the authors discuss a potential mechanistic basis for this behavior*?

This observation is made based on statistical calculations from the junctions of many, many cell clusters. Therefore we refrained from showing single cell clusters that may or may not be representative. We very much agree with the reviewer that this finding is very interesting and intriguing: we have now discussed a potential mechanistic basis in the manuscript text (Discussion section). To summarize: as previously reported by the Dufresne group and as we also showed in our manuscript, traction forces are higher along the outer periphery of the cluster, suggesting cell-cell forces derived from cell-matrix forces mainly at the cluster periphery. At the same time, as we reported in our manuscript, force transmission across a cell cluster is short-ranged due to the force scrambling and force anchoring mechanisms. On stiffer substrates, such mechanisms are more active, preventing force transmission to the center of the cluster, whereas on softer substrates, such mechanisms are less active, allowing force propagation through the cluster center. It is unclear why such mechanisms would be more active on stiffer compared to softer substrates, nevertheless, it has been shown by many reports that focal adhesions are larger (force anchoring mechanism) and actomyosin contractility is higher (force scrambling mechanism) in cells on stiffer substrates, and our laboratories have also shown that these differences extend multiple cell layers into a cell sheet (Ng et al., Journal of Cell Biology, 2012).

*It is not clear what the advantage of studying E-cadherin dynamics in such large clusters is. The method provides a relatively rough estimate of the force (14% precision as documented by the authors in a four-cell case, but probably less in larger clusters) whereas the correlation between force and E-cadherin concentration is a highly debated issue that requires exact force measurement. To be fully convincing, the positive correlation between force and E-cadherin and the increased correlation in growing and shrinking junctions, should be confirmed in cell doublets or linear triplets with exact intercellular force measurements. The time correlations displayed in*
Figure 7
*are particularly inconclusive. Could the authors either substantiate these findings with more convincing experiments or remove these experiments*?

We agree with the reviewers that coupling between force and cadherin recruitment is a complex question and that our correlation analysis is not robust enough to provide a conclusive statement. In order to not clutter this debate with yet another piece of uncertain data, we followed the reviewers’ suggestion and remove the experiments in Figure 7. However, we want to clarify that the 14% relative error in the force measurements are not specific to the FEM method. The force imbalance analysis in doublets and triplets offers a similar precision. This measurement uncertainty is predominantly associated with errors in the traction force measurements, which are the same for FEM or force imbalance analysis. We have stated this very explicitly in the manuscript, therefore the suggested reanalysis of temporal correlations on doublets would not add any certainty to this data.

*It does not seem correct to discuss a trend that is not substantiated by statistics (particularly compared to the statistics of the other experiments presented in the paper)*.

*Furthermore, the movies show strong protrusive activity, with contacts between cells in the plane of imaging, in contrast with the 'standard' geometry of adherens junctions. Yet, the determination of cell-cell transmission uses in plane traction force measurements and assumes traction forces perpendicular to the interface*.

*The observed correlation between changes in E-cadherin intensity and stresses might reflect this stronger protrusive activity. How do the authors cope with this*?

The reviewers are correct in that our setup is hardly capable of resolving whether or not the axis of a cadherin-cadherin junction lies within the focal plane or points out of that plane. In the latter case, the forces that are associated with such a junction would have the nature of shear stresses acting on the cadherin-cadherin link. The highly significant correlation between junctional forces and cadherin intensity that we found indeed only shows that the size of and the forces transmitted through cadherin adhesions are associated quantities. The analysis cannot tell however if these forces act in normal or shear mode on the molecular scale of a cadherin-cadherin linkage. To resolve such details, truly molecular experiments would be required.

*Minor comments*:

*The title does not seem to reflect the message of the paper; it suggests that the paper is about stress homeostasis, which is only one of the applications presented and which is not investigated in detail. We would suggest modifying the title to make clear that this is primarily a methods paper*.

We have modified the title: The revised title is “Mapping the dynamics of force transduction at cell-cell junctions of epithelial clusters.”

*Could the authors provide details on how the force extraction is exactly done? What is the code used? What are the required inputs? Can the calculation be performed with a normal computer and how long does it take*?

The whole computational part of this project has been performed in MATLAB. The solution of the thin plate model required functions of the “Partial Differential Equations” toolbox of MATLAB, which are listed in the Materials and methods. With respect to computation time, the most demanding step by far is the detection of bead displacements that are the input for the traction force calculations. It is an established method based on correlative template matching and takes on the order of several hours per movie. In contrast, solving the thin plate model is much faster and takes only several minutes. Each calculation step is not particularly demanding with respect to computational hardware. For instance, no large memory is required and the entire analysis can thus be performed on normal computers.

*Reviewer #1*:

*The magnitude of the total intercellular force per cell depends on the numbers of neighboring cells, whereas the magnitude of the traction force on the ECM does not. Why are cells and stress not shown here? How reliable are cell-cell force measurements by FEM in clusters containing more than 20 cells*?

As mentioned above, such observation is made from statistical calculations of many junctions of many, many cell clusters. Thus we feel it will be misleading to choose one cluster for representation. We have now included a discussion of the cluster size analyzable by our method in the Discussion section.

*Reviewer #2*:

*This manuscript is written in a very technical manner and the methods part at the beginning of the Results section but also later parts of the Results section are not easy to follow.*
Figure 1
*is over-crowded, it is not clear to me why the mathematical equations and the references are required here. In order to demonstrate the way the method works, it seems more helpful to proceed to*
Figure 3
*and to place*
Figure 2
*later or in the supplement*.

We sincerely apologize that the manuscript can be somewhat technical at times. The writing in the manuscript is the product of many discussions, edits and re-iterations between all the authors (our team is composed of a cancer biologist, a cell biologist, an electrical engineer, and a physicist), and we have strived with our best efforts to make the text accessible to a broad readership with both technical and non-technical backgrounds. We must admit that, without specifics, we are at a loss as to what could be changed to make the text even more accessible without comprising essential pieces of information or over-simplifying our methodology.

We have also very carefully considered the reviewer’s suggestion with respect to Figure 1. Our intention with Figure 1 is to provide, in one shot, an overview of the approaches that have been pursued so far to infer cell-cell forces, giving full credit to previous contributions (hence the references), as well as to introduce the critical issue of cluster topology. After a lengthy discussion among all authors where we tested several scenarios in which we would remove the figure, put it to a supplement, or split it up to make it less dense, none of us were satisfied with any of these options compared to the current Figure 1. We thus decided to leave Figure 1 as is. We appreciate the reviewer’s perspective on this and want by no means to come across as dismissing his/her advice. However, unless the editor feels removal of Figure 1 is requisite for publication we would prefer to leave it as is.

*The quantities analyzed in*
Figure 2
*are not defined well and it is not clear how the numbers for N and n correspond to each other*.

We apologize for the confusion, and have now clarified these quantities in the figure legend. N is the number of distinct cell-cell interfaces that we have analyzed. Each interface may have been analyzed over several frames of a time-lapse image sequence. Thus, from these N=263 interfaces we have gathered n=5750 measurements shown in the scatter plot. So on average each interface contributed ∼22 measurement points (some contributed more, some contributed less).

*In*
Figure 3
*it should be explained why the sign inverted traction forces are shown. I also note that the network representation is rather dense and maybe even redundant, because the connectivity information on the edges can also be read off from the nodes (I understand that this information is used later to analyze force distribution and I do not mind leaving it in if the authors find this essential, but it shows again that the presentation is very crowded). What seems to be missing from this representation, though, seems to be information if a given node has resulted from a cell division*.

We showed the sign inverted traction force to also demonstrate that there are cells that, despite generating a small traction force sum, are nevertheless under high cell-cell stresses. This is true and particularly interesting for the talin-kd cells in Figure 9 (previously Figure 10), which are inhibited in traction force generation but are set under cell-cell stresses by neighboring control cells. We do understand that the network plots can be rather crowded, but this is the most concise representation we could find that includes all required information. We like the reviewer’s suggestion of including information as to whether a node has resulted from a cell division; however, after much consideration we decided not to specifically highlight cells that result from cell divisions during a movie. One reason is that any two cells in a cluster may have originated from a cell division before the start of imaging. So the color code may falsely suggest that the highlighted cells are the only cells that may have resulted from cell division.

*In*
Figure 4*, the authors use a very special definition of strain energy (using only the displacement resulting from a single cell) without discussing alternatives or limitations*.

The chosen calculation scheme is a precise representation of the strain energy generated by a single cell in the cluster. Note that the straightforward integrating of the product of measured traction stresses and substrate strain over the footprint of a single cell would not give the correct result. This is due to the fact that elastic forces are long ranged. For a cell cluster this means that forces from one cell will cause substrate deformations outside of its own footprint, for instance underneath the footprint of a neighboring cell. The way we calculate cell strain energy treats these mechanical subtleties correctly and includes only the cell-own strains in the energy calculation.

*Regarding the length scale analysed in*
Figure 6*, I wonder if the authors effectively extract the persistence length of the cell-cell contour; the explanations of the meaning of these length scales are rather hard to follow. In general, the authors should rethink how to make the presentation of their results more accessible to the general reader*.

In our analysis, we correlated the absolute value of a cell-cell force vector that is transmitted through a small segment of the cell-cell interface with the E-cadherin-GFP intensity of that interface segment. Note that we take the absolute value, thus there is no directional information included in the correlation analysis. We did not extract the persistence length of the cell-cell contour.

*I also note that Title and Abstract might be further sharpened. The used of the word “active” is not clear on first reading*.

Following the suggestion of the reviewer, we have now revised the title and the Abstract.

*Although some details of the model are explained in large detail, the core of the method, namely the FEM-work, is not described in sufficient detail in the Materials and methods. Is the method implemented in the MATLAB PDE toolbox an optimization technique and if so, with which target function? What is the difference to the approach in monolayer stress microscopy, where one integrates the partial derivatives? Another issue is the use of an exponentially decaying Young's modulus between tight cluster mask and footprint boundary. Why is this complicated ad hoc procedure necessary? Can it happen that the Young's modulus has already decayed to a very small value at the cero stress boundary of the footprint and would this not lead to artifacts*?

The thin plate model is solved within MATLAB using PDE toolbox functions. The functions used are named in the supplement. No optimization scheme is applied; no target functions need to be defined. The underlying model is very similar to the one used in monolayer stress microscopy. The main difference is the fact that in our approach the cluster boundary is completely defined within the field of view. Therefore, in our approach a complete force balance equation for the cluster can be built up, which is impossible in monolayer stress microscopy. (Note that in monolayer stress microscopy, it is unknown which stresses are transmitted at the margins of the field of view occupied with cells.)

The extension of the tight cluster mask (which is the automatically segmented cell cluster footprint) towards the dilated cluster mask is performed to ensure that all significant traction forces are included in the cell-cell force calculations. A careful inspection of provided figures will show traction force vectors outside the footprint of the cells. There are several reasons for such traction forces outside the cluster footprint: the most important one is the limited spatial resolution of the traction force measurement. The presence of these traction force vectors outside the cell footprint presents a dilemma: One would ideally want to cut off the thin plate model at the tight cluster footprint, but then one would lose these “external” forces when setting up the force balance for the cluster. Our position on this dilemma is to include these forces by extending the tight cluster mask but on the other hand to give the mechanics “at least” a penalty with the distance to the detected cluster edge. Also note that the detection of the tight cluster mask is done automatically by image processing. This detection is associated with an uncertainty which for itself requires a certain lateral tolerance to make sure that significant traction forces are not excluded in the analysis.

*Regarding the cell-matrix force reconstruction, the authors use standard approaches and there is no real concern here, except that the method used (regularized FTTC) should be mentioned in the main text (at least when showing the traction maps in*
Figure 3*, which obviously use a cubic lattice) and that a number should be given for the spatial resolution (actually the bead density seems to be quite low)*.

In a standard procedure, the displacement field is detected at the actual bead positions and then interpolated on a regular cubic lattice (required for FTTC) with a lattice density that equals the average bead density. Since the traction forces are calculated on the same lattice, the shown density of traction forces resembles the average bead density. The spatial resolution is defined by the average bead density together with the size of the image template used during bead displacement detection which was 15-25pixels = 2.5-4.1um and is about, but not precisely, the bead density. We have added the method (“regularized FTTC”) used to the caption of Figure 3.

*What is the maximal bead displacement used in these experiments*?

The maximal bead displacement on 8kPa substrates is ∼16 pixels or ∼2.6µm, and the maximal bead displacement on 35kPa substrates is ∼6 pixels or ∼1µm. Below are two histograms showing the distribution of bead displacements pooled from 5 representative, independent cell clusters on 8kPa and 35kPa substrates, respectively.Author response image 2.

*How could the spatial resolution be improved in the future (leading to a smaller footprint region and to better results regarding the adhesions)? Such a discussion is important to understand the limitations of this approach and would also be helpful to understand the complicated procedure used to extract the sub-cellular length scale*.

Our analysis is mainly limited by the resolution of the traction force measurement. Any approaches to improve TFM will also improve the cell-cell force measurements. One way is to increase bead density by using differently colored beads. Using higher magnification objectives will also improves the TFM measurements but that comes with the drawback of a reduced field of view, which means that stitching/tiling of overlapping images will be required to capture cell clusters of the size we studied. The numerical schemes for TFM per se may also offer room for improvement, for instance via the regularization scheme, which can have a significant influence on the effective resolution of TFM. We have now explicitly mentioned in the Discussion section that the main source of uncertainty in our approach is the TFM input.

*Why do global patterns of force distribution arise when at the same time the authors argue that force transmission is locally scrambled by cell-matrix contacts and actomyosin contractility? In fact these results seem to be contradictory, because stress localization to the rim is exactly what one expects from thin plate theory with long-ranged forces; because the authors use this assumption in their method, it is not clear how consistent their conclusions are in regard to the method used*.

The reviewer suggests that the thin plate model tends to introduce artificial stresses at the plate boundary and that this could be the reason for the global patterns of stress distributions that we find in clusters. Here we respectfully disagree with the reviewer. For instance we find differences in global stress distributions between clusters on 8 vs 35 kPa. If the findings on 35 kPa were due to a systematic error, one would expect to find the same pattern on 8kPa substrates. Furthermore, the reviewer should note that the interfaces of cells at the cluster boundary are usually fairly perpendicular to the cluster boundary. Thus, even if there were artifacts at the boundary, they would impact the cell-cell interface only along the first segments. Note that stresses are integrated along the whole length of the interface of which only a small part is close to the boundary.

The global pattern of force distribution we found does not contradict the suggested short range force transmission. Cells are able to sense their environment and respond appropriately. Therefore it can be expected that cells at the cluster boundary behave differently compared to cells within the cluster. There is no long-range mechanical cluster-boundary-to-cluster-inside communication needed to build up this heterogeneity in cell-cell stresses.